# FLASH INFERENCE: NEAR LINEAR TIME INFERENCE FOR LONG CONVOLUTION SEQUENCE MODELS AND BEYOND

**Costin-Andrei Oncescu, Sanket Purandare, Stratos Idreos & Sham Kakade**
Harvard University
`{concescu,sanketpurandare}@g.harvard.edu,`
`{stratos,sham}@seas.harvard.edu`

## ABSTRACT

While transformers have been at the core of most recent advancements in sequence generative models, their computational cost remains quadratic in sequence length. Several subquadratic architectures have been proposed to address this computational issue. Some of them, including long convolution sequence models (LCSMs), address this issue at training time but remain quadratic during inference. We propose a method for speeding up LCSMs' exact inference to quasilinear time, identify the key properties that make this possible, and propose a general framework that exploits these. Our approach, inspired by previous work on relaxed polynomial interpolation, is based on a tiling which helps decrease memory movement and share computation. It has the added benefit of allowing for almost complete parallelization across layers of the position-mixing part of the architecture. Empirically, we provide a proof of concept implementation for Hyena, which gets up to $7.8\times$ end-to-end improvement over standard inference by improving up to $110\times$ within the position-mixing part.

## 1 INTRODUCTION

A lot of recent progress in deep learning, particularly in the form of large language models (LLMs) has been driven by the transformer architecture (Vaswani et al., 2017). While these models have great quality, it comes at a computation cost which scales quadratically in sequence length - both during training and inference. This can become prohibitive for very long contexts and as such a number of alternative architectures with better computational scaling in context length have been proposed (Gu & Dao, 2023; Poli et al., 2023; Fu et al., 2024). Although most of these works have improved computational efficiency for training, some still scale quadratically in sequence length when it comes to inference, thus not improving asymptotically over transformers.

In this work, we propose a framework for optimizing inference efficiency for a general class of such models. As a case study, which inspired the method, we focus on long convolution sequence models (LCSMs) (Poli et al., 2023; Fu et al., 2022; Romero et al., 2021b; Li et al., 2022; Karami & Ghodsi, 2024; Fu et al., 2023a; Romero et al., 2021a). However, our approach is not limited to LCSMs alone and we identify the properties that allow for such inference speedups in hope to guide the design of future architectures.

In the particular case of LCSMs, the building block of the architecture is that of convolving the input sequence with a sequence-length long, (*potentially* underparameterized) filter. If we let $L$ be the sequence length (e.g. number of tokens in the case of LLMs), then a naive implementation of convolution during training would take $\Omega(L^2)$ FLOPs, but one can employ FFT to bring that down to $O(L \log L)$. During inference, FFT cannot be used directly since the whole input sequence is not known ahead of time, but rather incrementally computed. Because of this, the naive inference approach goes up to $\Omega(L^2)$ - this is the apparent cost of moving from a *static* input to a *dynamic* one.

It turns out that in fact, at least in the case of "dynamic FFT", one can obtain $O(L \log^2 L)$ time complexity by using van der Hoeven (1997)'s result on relaxed polynomial convolution. Crucially,

this result works by a direct reduction to several applications of FFT. This allows us to phrase a more general framework for turning training-efficient architectures to inference-efficient ones.

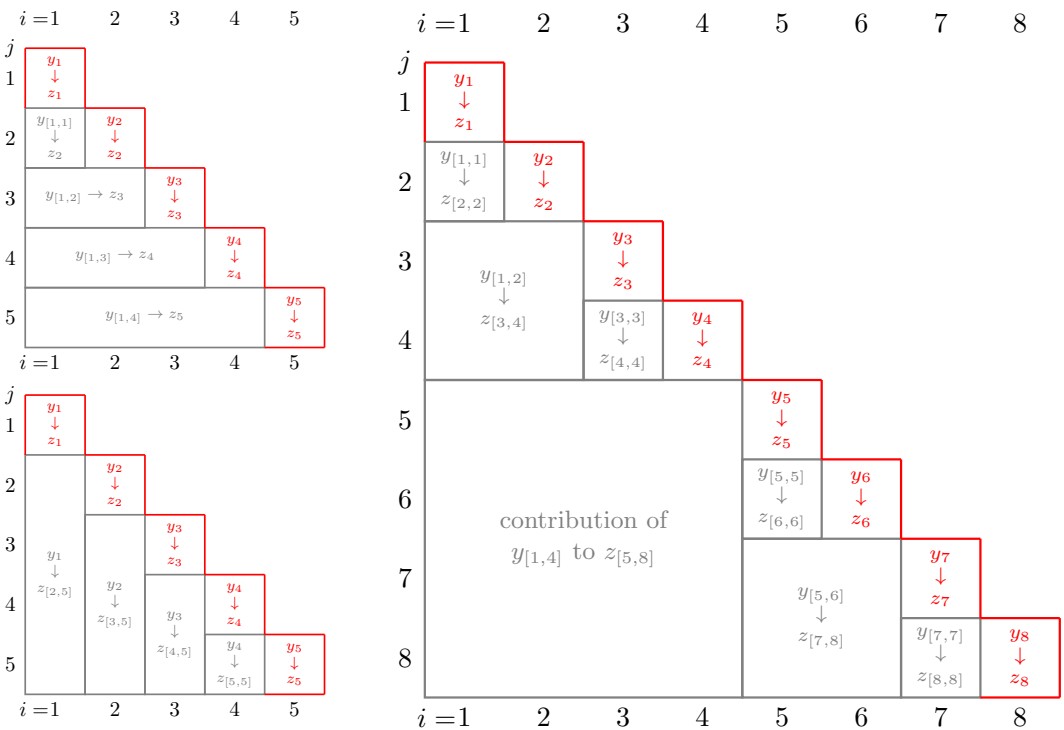

Figure 1: Cell $(i, j)$ corresponds to the contribution of $y_i$ to $z_j$. To compute $z_j$, all its line of contributions should've been accounted for. Because of the autoregressive nature of inference, one only has access to $y_i$ after $z_{i-1}$ has been computed. *(Left Top)* represents the standard lazy approach, *(Left Bottom)* represents the eager approach, and *(Right)* represents our suggested method.

Our main contributions are:

(1) Proposing the first quasilinear-in-sequence-length $O(L \log^2 L)$, *exact* inference algorithm for LCSMs. This is in contrast to previous work of (Massaroli et al., 2024) who performs approximate inference.

(2) Identifying the main features of LCSMs that allow for faster inference to propose a more general framework.

(3) Highlighting and exploiting the potential for dealing with most workload of different layers in parallel, rather than having to wait for all computation of a layer to finish before moving on to subsequent layers.

(4) Saving up on data movement thanks to the tiling approach used.We show how to save factors of 2 in several places, including activation storage, when the convolutional filters are data-independent. However, our method extends to any causal, data-dependent filters.

(5) Our framework provides an $O(L \log^2 L)$ inference algorithm which, when run empirically, yields up to $7.8\times$ end-to-end time-efficiency improvement; and up to $110\times$ on the long convolution part.

## 2    SETTING AND RELATED WORK

Recent efforts have focused on designing more efficient alternatives to transformers (Tay et al., 2022), with state space models (SSMs) (Gu et al., 2021a; Gu & Dao, 2023) and convolution-based models (Fu et al., 2022; Poli et al., 2023) emerging as notable contenders. In this work, we focus on the latter simply because we leverage the convoluional view directly. Nevertheless, it is worth

mentioning that there is a one-to-one mapping between linear-time invariant (LTI) SSMs and LCSMs and consequently, our findings apply to LTI SSMs as well (since they represent the same models). While LCSMs inherently assume sequence-long convolutions, they can correspond to LTI SSMs of varying state dimensions (up to filter length). Using the recurrent formulation of SSMs for inference incurs costs that scale with the state dimension, which can become prohibitive for larger dimensions. In contrast, our method exclusively leverages the convolutional perspective and has no dependency on filter properties. An extended contextual discussion is available in Appendix A.

## 2.1 NOTATIONS AND DEFINITIONS

In the most general form, we consider architectures obtained by stacking several position-mixing layers potentially interleaved with element-wise, feature-mixing modules, partially following Arora et al. (2023)'s notation. Put formally, let $\mathrm{block}^1, \ldots \mathrm{block}^M : \mathbb{R}^D \to \mathbb{R}^D$ be feature-mixing modules and $\mathrm{mixer}^1 \ldots \mathrm{mixer}^M : \mathbb{R}^{L \times D} \to \mathbb{R}^{L \times D}$ be position-mixing modules (we use superscripts throughout to denote layers). One can think of the blocks as being combinations of MLPs, layernorms, skip connections and other modules of similar purpose and of mixers as playing the role of multi-head attention (MHA) in a transformer - the part where embeddings at different positions interact with each other. Assume these are all learnt from some well-defined parameterized classes. Here, $M$ is the number of layers, $D$ is the used embedding dimension and $L$ can be any sequence length (unless positional embeddings are used, case in which it can only take values up to some $L_{max}$). We define the activations $a^0, \ldots, a^M : \mathbb{R}^{L \times D} \to \mathbb{R}^{L \times D}$ at level $1 \leq \ell \leq M$ and position $1 \leq i \leq L$ as $a^\ell(x)_i \triangleq \mathrm{block}^\ell(\mathrm{mixer}^\ell(a^{\ell-1}(x))_i)$ where $a^0(x) = x$ represents the input embeddings ($L$ $D$-dimensional tokens) and the model output at position $i$ is read from the last layer $a^M(x)_i$.

We focus on autoregressive sequence models where for an input sequence $x_1, \ldots x_p$, one generates the $(p+1)^{\text{th}}$ token by sampling from a distribution given by $a^M(x)_p$ (usually via a linear map). For this to be well-defined, we constrain (by construction of $\mathrm{mixers}$) the models to be causal, that is, $a^\ell(x)_i$ is a function of only $x_{[1,i]}$ or, equivalently, $\mathrm{mixer}^\ell(y)_i$ is a function of $y_{[1..i]}$.

We use $\odot$ to denote the element-wise (Hadamard) product: for two matrices $A, B \in \mathbb{R}^{N \times M}$, $A \odot B \in \mathbb{R}^{N \times M}$ is given by $(A \odot B)_{i,j} = A_{i,j} \cdot B_{i,j}$. We call a function $f : \mathcal{X}^* \to \mathcal{X}$ associative, if $f(x_1, x_2, \ldots x_p) = f(f(x_1, \ldots x_i), f(x_{i+1}, \ldots x_p))$ for any $1 \leq i < p$ and $x_1 \ldots x_p \in \mathcal{X}$. Finally, we define the time complexity of an algorithm as the number of floating-point operations (FLOPs) it executes - this is a notion independent of hardware.

## 2.2 SELF-ATTENTION

Self-attention is the building block of transformers (Vaswani et al., 2017). In its simplest form, it implements a $\mathrm{mixer}$ by using three projection matrices $Q, K, V \in \mathbb{R}^{D \times D}$ to obtain three sequences of vectors $q, k, v \in \mathbb{R}^{L \times D}$ by projecting the input $y \in \mathbb{R}^{L \times D}$ via $q = yQ, k = yK, v = yV$ - these are called queries, keys and values, respectively. The (causal) self-attention operator $attention :$ $\mathbb{R}^{L \times D} \to \mathbb{R}^{L \times D}$ is then given by:

$$attention(y)_j \triangleq \mathrm{mixer}(y)_j = \frac{\sum_{i=1}^j v_j \cdot e^{\langle q_j, k_i \rangle}}{\sum_{i=1}^j e^{\langle q_j, k_i \rangle}} = (v_{[1,j]})^\top \mathrm{softmax}(k_{[1,j]} q_j) \qquad (1)$$

which represents, at position $j$, the average of the previous value vectors $v_{[1,j]}$ exponentially-weighted by how well the values' corresponding keys match $j^{\text{th}}$ query. Put differently, the bigger $\langle q_i, k_j \rangle$ is, the more $j^{\text{th}}$ output attends to $i^{\text{th}}$ input. Note that both in the above notation, as well as in general, we assume any 1-dimensional tensor to be a column vector (for example $q_j \in \mathbb{R}^{D \times 1}$). In the transformer architecture this is how one "head" operates and each embedding is normally split into several such heads - this is called multihead attention (MHA). If we take the causality away (that is, the $i \leq j$) for simplicity, one could think of attention as being $\mathrm{mixer}(y) = \mathrm{softmax}(qk^\top)v$ where $\mathrm{softmax}$ is computed along the rows of what is called the attention matrix: $qk^\top$.

## 2.3 LONG CONVOLUTION SEQUENCE MODELS (LCSMs)

LCSMs (Poli et al., 2023; Li et al., 2022; Gaido et al., 2024) work by creating a SISO (single-input-single-output) primitive to map an input sequence $y \in \mathbb{R}^L$ to $z \in \mathbb{R}^L$ via $z_t \triangleq \sum_{i=1}^t y_i \cdot \rho_{t-i}$ where

$\rho$ is a (possibly infinite, and of length at least $L$) convolution filter which is *often* parameterized by a smaller latent space $\Theta$: $\rho = f(\theta)$ where $\theta \in \Theta, \dim \Theta \ll L$ is learnt. These convolution primitives operate on independent axes of the hidden dimensions to create a positional mixer: $\mathrm{mixer}(y)_t \triangleq \sum_{i=1}^{t} y_i \odot \rho_{t-i} \in \mathbb{R}^D$. This assumes the filters to be independent, but shared ones are possible as well (as is the case for multi-head Hyena (Massaroli et al., 2024)).

For example, the Hyena architecture (Poli et al., 2023) maps back to our setup when mixers are defined as above and the block functions, depending on the layer, are either MLPs or gates - that is, element-wise multiplications with a projection of activations at same position, but lower level. We do not focus on the details of the blocks since they all involve some $D \times D$ matrix-vector multiplication that is performed once for every layer and position $1 \le i \le L$ and thus scale as $\Theta(LD^2)$ per layer - that is, linearly in context length.

### 2.3.1 THE INFERENCE PROBLEM

Whereas during training, the convolutions can be performed in $O(L \log L)$ time via FFT as the whole of $y$ is known beforehand, during inference this is non-trivial. Suppose there is a prompt of size $P$ and we want to generate $L - P$ more tokens. The bottleneck problem can be understood as computing, for every layer and every dimension, the below:

$$z_t \triangleq \sum_{i=1}^{t} y_i \cdot \rho_{t-i} \tag{2}$$

for all $1 \le t \le L$ . To pre-fill the first $P$ values $z_{[1..p]}$ at all levels of activations, since the first $P$ inputs are known, one can perform FFT as they would normally do during train-time (incurring an $O(P \log P)$ time cost), but past $P^{\mathrm{th}}$ position it is important to note that $y_i$ is not available before computing $z_{i-1}$ - this is where *static* FFT breaks. Since dealing with the prompt can be done easily (Massaroli et al., 2024, Lemma 2.1) by essentially filling in all contributions of $y_{[1..P]}$ to $z_{[1..L]}$ and then forgetting the prompt ever existed, we henceforth assume $P = 0$.

### 2.3.2 PREVIOUS WORK ON EFFICIENT INFERENCE

Speeding up Equation 2 from the naive $\Omega(L^2)$ is the object of study of Massaroli et al. (2024): they optimize the process by finding a *low-dimensional* LTI SSM whose equivalent convolution filter closely resembles $\rho$, thus getting an RNN-like problem to simulate. If the learnt SSM has size $D'$, then this yields an $\Theta(LDD')$ algorithm for generating $L$ tokens. This has the significant added benefit of not needing to store the activations (or even inputs) thus far which makes it memory efficient and practically time-efficient.

The downside, however, is that by definition this will only be an approximation of the learned filter. More importantly, this approximation represents a projection to a potentially different space of models, making the end-to-end procedure practically a different training procedure for a *low-dimensional* LTI SSM model. Furthermore, this approach assumes the filter $\rho$ is data-independent - this is needed in order to undergo an expensive distillation procedure as a precomputation step. This can be qualitatively limiting as shown in Arora et al. (2023). While we do store all activations, our proposed framework exactly simulates the architecture and data-dependent filters can also be accommodated.

## 3 FAST LCSM INFERENCE

In this section we describe our method for the case of LCSMs. That is, we still work with models as described in Section 2.1, but we further assume the mixers to be convolution-based, as described in 2.3. Hence, for every $\mathrm{mixer}^{\ell} \in \{\mathrm{mixer}, \ldots \mathrm{mixer}^M\}$, there exists a filter $\rho^{\ell} \in R^{L \times D}$ such that

$$\mathrm{mixer}^{\ell}(y)_t = \sum_{i=1}^{t} y_i \odot \rho_{t-i}^{\ell}$$

We assume here, for simplicity, that $\rho$ is data-independent and part of the model, as is the case for most existing architectures. However, we discuss in Appendix C how our method can be extended

to data-dependent filters (by paying a factor of 2). We also assume $L = 2^P$ for some integer $P$ for simplicity - one can always round $L$ up to the closest power of 2 without changing it by more than a factor of 2 and thus keeping the asymptotics of all our analysis unchanged.

## 3.1 The Proposed Algorithm

We derive inspiration from the the work of van der Hoeven (1997) regarding relaxed polynomial interpolation - they propose a fix to dealing with *dynamic* structure of Eq. 2 to achieve an overall $O(L \log^2 L)$ time complexity.

### 3.1.1 Relaxed Polynomial Interpolation

Consider Eq. 2:

$$z_t \triangleq \sum_{i=1}^{t} y_i \cdot \rho_{t-i}$$

The problem of relaxed polynomial interpolation can be thought of as having to compute $z_t$ for every $1 \leq t \leq L$ with the further constraint that $y_t$ and $\rho_t$ are only made available once $z_{t-1}$ has been outputted. While this general setting is covered in Appendix C, here we focus on the case that all of $\rho$ is known ahead of time and only $y$ is incrementally revealed. Whereas one could simply use the more general approach of (van der Hoeven, 1997), we can further take advantage of $\rho$ being known ahead of time to get a twice-faster and slightly simpler algorithm. We present this both for brevity and because existing architectures have data-independent $\rho$'s so this is the more efficient choice for them (and thus, for practical purposes). However, (Arora et al., 2023; Karami & Ghodsi, 2024) show that data-dependent filters can improve quality but finding a good causal architecture remains open.

The main approaches to relaxed polynomial interpolation, as identified in (Van der Hoeven, 2002), are:

- The lazy approach: for every $t \in \{1 \ldots L\}$, one computes $z_t$ by simply applying the formula. It takes $O(t)$ FLOPs to do so and the overall complexity is $O(L^2)$. This approach can be thought of as the naive approach which only performs work when it is strictly needed.

- At the opposite end, there is the eager (or zealous) approach: as soon as a new value of $y$ or $\rho$ becomes available, one accounts for all its contributions to $z$. Since we assumed the entire $\rho$ is known beforehand, this translates to the following:

  After computing $z_{t-1}$, $y_t$ becomes available, so for every $t \leq i \leq L$, we increase $z_i$ by $y_t \cdot \rho_{i-t}$. $z_t$ is now fully computed so we proceed to the next iteration.

  Hence, the eager approach can be thought of as performing work as soon as it can be performed. Here, $t^{\text{th}}$ iteration takes $O(L - t)$ so we still have an overall complexity of $O(L^2)$.

- Relaxed approaches: these approaches perform some but not all available work ahead of time - their advantage stems from grouping contributions (i.e. accounting collectively for groups of pairs $y_i \cdot \rho_j$) in such a way that fast convolution methods (such as FFT) can be employed for a speedup.

First, let us define the sort of contribution grouping that we consider:

**Definition.** *For any $1 \leq l \leq r \leq l' \leq r' \leq L$ we let $\tau(y, [l, r], \rho, [l', r'])$ represent the aggregated contribution of all $y_{[l,r]}$ to all of $z_{[l',r']}$. That is, for all $l' \leq t \leq r'$, we have:*

$$\tau(y, [l, r], \rho, [l', r'])_t \triangleq \sum_{i=l}^{r} y_i \cdot \rho_{t-i} \tag{3}$$

The employment of FFT for speeding up the computation of $\tau(y, [l, r], \rho, [l', r'])$ can then be achieved following the next lemma:

**Lemma 1.** *Let $1 \leq l \leq r \leq l' \leq r' \leq L$ represent ranges of lengths $L_1 = r - l + 1$ and $L_2 = r' - l' + 1$ of $y$ and $z$, respectively. There exists an FFT-based algorithm running in $O(L_1 + L_2)$ space and $O((L_1 + L_2) \log(L_1 + L_2))$ time complexity that, given access to $y_{[l,r]}$, computes all the elements of $\tau(y, [l, r], \rho, [l', r'])_{[l',r']}$ - that is, all the aggregated contributions of $y_{[l,r]}$ to all of $z_{[l',r']}$.*

What this lemma says is that we can compute efficiently the contribution of a range of inputs to a range of outputs, where the computational cost is asymptotically given by the larger of the ranges (this is because $O(L_1 + L_2) = O(\max(L_1, L_2))$). Furthermore, observe that we assumed $r \leq l'$: this is because in order to account for all these contributions at once, we need to have all of $y_{[l,r]}$ available which means that all of $z_{[1,r-1]}$ was computed, so by that time we should have already accounted for the contributions of $y_{[l,r]}$ to all of $z_{[1,r-1]}$; hence, there is no point to (re)computing any of these.

The lazy, eager, and our version of a relaxed approach are depicted in Figure 1 under the form of tilings of the contribution space. Note that the lazy and eager approaches can still be thought of as grouping contribution accounting, but do so along thin strips: the contribution of a long range of $y$'s to one element of $z$ in the lazy approach and that of a singe element of $y$ to a long range of $z$'s in the eager approach. The key to our relaxed tiling is using more balanced tiles since, as noted, the cost of a tile is given by the larger side (which is very expensive for thin tiles). The tiles still have to respect $r \leq l'$ and one needs to ensure that the whole line of contribution to some $z_t$ has been dealt with by the time we return the value of $z_t$ - this is indeed the case for our tiling.

---

**Algorithm 1** Fast Relaxed Polynomial Interpolation

---

**Require:** filter $\rho \in \mathbb{R}^L$ and $y_1$
1: Initialize $z \in \mathbb{R}^L$ with zeros
2: **for** $i \leftarrow 1$ **to** $L - 1$ **do**
3:     $U \leftarrow$ maximum power of 2 that divides $i$ # the side of the $i^{\text{th}}$ gray tile
4:     $z_i \mathrel{+}= y_i * \rho_0$ # finalize $z_i$ by accounting for newly available $y_i$ - red cell
5:     # account for the contribution of $y_{[i-U+1,i]}$ to $z_{[i+1,i+U]}$ - gray tile
6:     $z_{[i+1,i+U]} \mathrel{+}= \tau(y, [i - U + 1, i], \rho, [i + 1, i + U])$ # $\tau$ is defined as in Eq 3
7:     return $z_i$ # $z_i$ has been computed
8:     $unlock(y_{i+1})$ # $y_{i+1}$ becomes available
9: **end for**
10: $z_L \mathrel{+}= y_L * \rho_0$
11: return $z_L$ # $z_L$ has been computed

---

While the lazy and eager algorithms have been described, one needs to make explicit the order of processing the tiles in the relaxed setting since we still cannot use $y_t$ before $z_{t-1}$ has been computed. This is shown in Algorithm 1: *return* marks a value of $z$ having been computed and *unlock* represents a value of $y$ becoming available. At the beginning of iteration $i$ it always holds that $z_i = \sum_{j=1}^{i-1} y_j \cdot \rho_{i-j}$. We start the iteration by completing this with the freshly unlocked $y_i \cdot \rho_0$ at line 4 - this corresponds to the red tiles of Figure 1. Following this step $z_i$ is ready to be returned. Then, we account for the gray tile that has just been unlocked (line 6) - this tile corresponds to the contribution of $y_{[i-U+1,i]}$ to $z_{[i+1,i+U]}$ where $U$ is the side of the tile, namely the largest power of 2 dividing $i$. Finally we return $z_i$ as being in its final form and get access to $y_{i+1}$ - we could have done this before dealing with the gray tile as well. Note that the relative ordering of the red and gray lines does not matter within an iteration, but it is important to finish up the first red and gray tiles before second ones, the second ones before the third ones and so on.

**Proposition 1.** *When $L = 2^P$, Algorithm 1 performs $2^{P-1-q}$ calls to $\tau$ on ranges of length $2^q$ for every $q \in \{0 \ldots P - 1\}$. If we base $\tau$'s implementation on Lemma 1, this entails a total time complexity of $O(L \log^2 L)$. Since the calls to $\tau$ are sequential and outputs are not stored, the peak memory usage is given by the largest call to $\tau$ and remains $O(L)$.*

### 3.1.2 PERFORMING LCSM INFERENCE VIA FAST RELAXED POLYNOMIAL INTERPOLATION

We can now use Algorithm 1 to get an efficient inference algorithm for LCSMs, but to do so we need to be careful about data-dependency. First, for brevity, in our previous notation of activations, disregard the dependency on input $x$ - that is, use $a_i^\ell$ instead of $a^\ell(x)_i$ - since we are only going to work with one input $x$ (the one we are generating). Furthermore, denote the intermediate mixer

computation by $b \in \mathbb{R}^{M \times L \times D}$ as follows:

$$b_i^\ell \triangleq \text{mixer}^\ell(a^{\ell-1})_i = \sum_{k=0}^i a_k^{\ell-1} \odot \rho_{i-k}^\ell \tag{4}$$

$$a_i^\ell \triangleq \text{block}^\ell(b_i^\ell) \tag{5}$$

for any $1 \le \ell \le M$, where $a_i^0$ is the embedding of $i^{\text{th}}$ token. Whenever one has generated the tokens $x_{[1,i]}$ up to position $i$ or, equivalently, $a_{[1,i]}^0$, the whole of $a_{[1,i-1]}^{[0,M]}$ will have been computed (since $a_{[1,i-1]}^M$ is needed to sample $x_{[1,i-1]}$). Caching these up is the equivalent of a KV-cache in transformers and is simply the natural way to not repeat work when autoregressively sampling. To generate the next token $x_{i+1} = a_{i+1}^0$, we want to fill in the values of $a_i^{[1,M]}$. To do so, as per Eq. 4, we need to compute $b_i^\ell$ which is practically, for each level $\ell$, and within each dimension $1 \le c \le D$, a relaxed interpolation between $a_{:,c}^{\ell-1}$ (playing the role of $y$) and $\rho_{:,c}^\ell$. Further applying a $\text{block}^\ell$ to $b^\ell$, one gets $a^\ell$. Thus, we can pipeline several instances of Algorithm 1 sequentially across layers.

The resulting algorithm is illustrated in Algorithm 2. The outer loop (line 3) iterates through positions $i$: at each iteration, one computes the activations at position $i$ across all layers, does some (limited) eager work and samples next token. To do so, the inner loop (line 5) iterates through the several mixer-layers. It first accounts for the red cells in the diagram, that is, the direct dependency on previous layer's activations at position $i$ (line 7), thus finalizing $b_i^\ell$. After computing the activation at position $i$ (line 8), it also accounts for the contribution of the gray tile that has just been unlocked (line 10) at the current layer. We reuse the notation from previous section for $\tau$ to be the same as before but within each of the $D$ dimensions - that is:

$$\tau(y, [l, r], \rho, [l', r'])_t \triangleq \sum_{i=l}^r y_i \odot \rho_{t-i}$$

for all $l' \le t \le r'$. The accounting works by considering (in-place) the influence of last $U$ positions of $a$ to next $U$ ones of $b$.

---

**Algorithm 2** Flash Inference for LCSMs

---

**Require:** filter $\rho \in \mathbb{R}^{M \times L \times D}$, $\text{block}^{[1,M]}$, first token $a_1^0$ and sampler
 1: **Output:** All activations $a^0, \ldots, a^M \in \mathbb{R}^{L \times D}$ obtained by autoregressively sampling
 2: Initialize $b \in \mathbb{R}^{M \times L \times D}$ to zeros
 3: **for** $i \leftarrow 1$ **to** $L - 1$ **do**
 4:     $U \leftarrow$ maximum power of 2 that divides $i$ # the side of the $i^{\text{th}}$ gray tile
 5:     **for** $\ell \leftarrow 1$ **to** $M$ **do**
 6:         # account for the contribution of $a_i^{\ell-1}$ to $b_i^\ell$ - red cell
 7:         $b_i^\ell \mathrel{+}= a_i^{\ell-1} \odot \rho_0^\ell$
 8:         $a_i^\ell = \text{block}^\ell(b_i^\ell)$
 9:         # account for the contribution of $a_{[i-U+1,i]}^{\ell-1}$ to $b_{[i+1,i+U]}^\ell$ - gray tile
10:         $b_{[i+1,i+U]}^\ell \mathrel{+}= \tau(a^{\ell-1}, [i-U+1, i], \rho^\ell, [i+1, i+U])$
11:     **end for**
12:     # generate next token based on the output of last layer at position $i$
13:     $a_{i+1}^0 = \text{sampler}(a_i^M)$
14: **end for**

---

As far as performance goes, Algorithm 2 practically calls Algorithm 1 $MD$ times (within each layer and within each of the $D$ dimensions of embeddings). Proposition 2 covers its complexity analysis:

**Proposition 2.** *The overall time complexity of Algorithm 2 is $O(MDL \log^2 L)$ for the mixer-part plus $LM$ block calls, to generate $L = 2^P$ tokens. Furthermore, for each $0 \le q \le P - 1$, there are $MD2^{P-1-q}$ calls to $\tau$ on ranges of length $2^q$. The overall amount of memory it takes to store the activations is $O(MLD)$ (as is the case for the naive lazy approach) and the peak memory usage does not change the asymptotics.*

### 3.2 ACROSS-LAYER PARALLELIZATION

One important feature of the proposed method is that it allows for higher parallelization across layers. In particular, the gray lines can be taken out of the inner loop and performed in parallel across all layers at once, just after the loop - this is because all their inputs and outputs are disjoint. However, the red lines need to be performed sequentially since $a_i^\ell$ is a function of $b_i^\ell$ which is a function of $a_i^{\ell-1}$ for every $\ell$.

Note that this optimization can be applied to the eager and lazy approaches as well (provided that the red tiles are accounted for separately from the gray ones). However, the benefit of extra parallelization is practically relevant only in settings that are not memory-bandwidth bound - this is the case for smaller tiles which represent a significant amount of the tiles in our method, but an insignificant amount in the eager and lazy approaches. This is further discussed in Appendix F.

### 3.3 MEMORY CONSIDERATIONS

As follows from Proposition 1, the overall size of the inputs to (and outputs of) $\tau$ in the proposed algorithm is only $O(MDL \log L)$ - that is, we access on average activation values at $\log L$ positions per iteration as opposed to the naive implementations (such as lazy and eager approaches) that access an average of $\Omega(L)$ positions. This directly translates to significant data movement improvements.

A simple optimization of static memory is to never store $b$, but rather use $a_i^\ell$ to store $b_i^\ell$ until hitting line 8: this will work because from that point onwards $b_i^\ell$ is never used again and $a_i^\ell$ was not needed (or available) thus far. Hence, the static memory overhead is minimal: at least in the LCSM case, one directly aggregates contributions to the same activation tensor, thus not using any extra storage (on top of the inherent $MLD$ it takes to store the activations).

The only extra cost we pay is in the peak memory usage which is given by the largest tile - this is $O(MLD)$, but can be dropped to $O(LD)$ at essentially no cost by dropping parallelization for large tiles, as further discussed in Appendix F.

Lastly, if one is restricted to only using one GPU and the whole tensor of $M \times L \times D$ cannot fit it, it is possible to drop the requirement to only storing $M \times (L/2) \times D$ activations by dropping parallelization in the largest tile (which was alluded above). This, together with a more granular analysis of memory usage is further discussed in Appendices E and Appendix F.

### 3.4 GENERALIZATION

We propose a framework called Flash Inference to generalize our fast LCSM inference algorithm, covered in Appendix B due to lack of space. Crucially, we identify the main properties that are being exploited by Algorithm 2: in order to apply our tiling one needs the mixers to be contribution-based (that is, $\text{mixer}(y)_i$ is some aggregation of contributions of $y_{j \leq i}$ to position $i$) and these contributions to only depend on data available at step $j$. The latter is the reason we cannot apply Flash Inference to transformers since $\langle q_i, k_j \rangle$ depends on $q_i$ which depends on $y_i$.

## 4 EXPERIMENTS

We performed two kinds of experiment: one synthetic and one on the Hyena architecture (Poli et al., 2023). The synthetic setup defines all blocks to be MLPs with a hidden dimension of $2D$ and GELU activation; it also simply sets $a_{i+1}^0$ as $a_i^M$ plus some noise to avoid dependency on vocabulary size since that is out of the scope of our framework and will be negligible at scale. Note that this can be viewed as a sampler: a function from logits at the last layer and previous position to the next token's embedding. In both settings the weights are initialized to random noise since it does not affect the runtime - this avoid unnecessarily training a new model for each hyperparameter choice.

In terms of notation, we introduce the batch dimension $B$ and keep $M, D, L$ refer to the number of layers, embedding dimension and number of tokens generated, respectively. We use $U$ to refer to the square-tile length, as per line 4. We sweep over $B \in \{1, 2, 4, 8\}, M \in \{18, 36\}, D \in \{256, 768\}$ and generate tokens up to the greatest power of 2 that fits the memory. All results are obtained

by averaging over $4$ runs following $2$ runs of warm-up. We evaluate our approach on on the latest NVIDIA H100 and A100 GPUs. (Results for all the parameter sweeps in supplementary material).

For our baselines, (1) we consider the eager and lazy approaches described in Section 3.1, depicted in 1. We also consider their implementation exploiting our observation regarding parallelization across layers (3.2) (we denote the the parallel versions simply as lazy and eager).

Our Flash Inference framework explored various implementations of $\tau$, covered in Section 4.2. Our best method is a *Hybrid* that dynamically chooses the optimal implementation of $\tau$ depending on $(B, D, M, U)$.

## 4.1 INTEGRATING FLASH INFERENCE IN REAL WORLD SETTING (HYENA ARCHITECTURE)

Flash Inference significantly speeds-up the end-to-end inference by up to $7.8\times$ and the convolution-based mixer component of the Hyena architecture by $110\times$ compared to the baselines as shown in Figures 2a and 2b. Figure 2c shows the per token response time of Hybrid and baselines. Hybrid shows low variance in per-token time except at the tokens positions where large tiles are computed. For a given sequence length of $L$, we have $L/2$ positions that use tile size 1, $L/4$ positions using tile size 2, and so on. That is, $93.75\%$ of tokens use a tile size $U \leq 8$. Hence the spikes occur rarely.

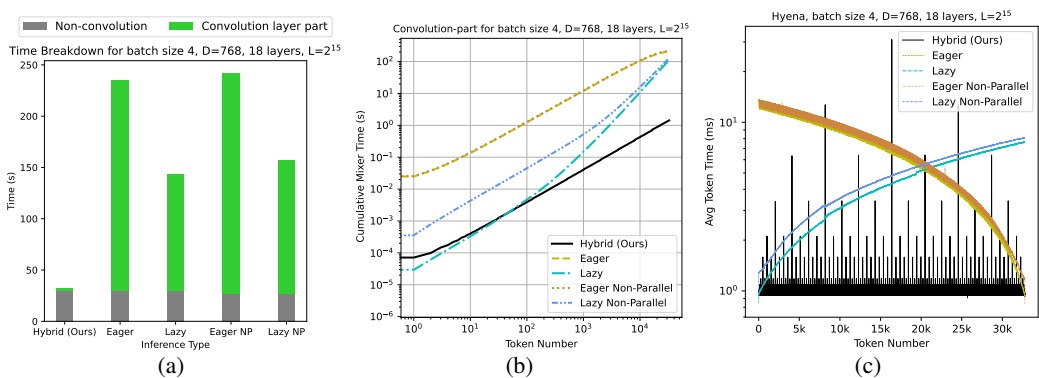

Figure 2: Real world Hyena experiments: (a) End-to-end inference time breakdown shows Hybird provides $4.8\times$ speed-up over optimized baselines (b) Cumulative mixer time of Hybrid scales $90\times$ better (c) Hybrid shows low variance in per-token response time except at the tokens positions where large tiles are computed.

## 4.2 $\tau$ IMPLEMENTATIONS

Algorithm 2 assumes an implementation of primitive $\tau$ that accounts for the contribution of a range of inputs to a range of outputs of a convolution. We considered 7 different implementations but, here, we only present results of the ones on the Pareto Frontier - that is, those optimal for at least some $(B, N, D, U)$ setting (the others are covered in Appendix D.1). There are $4$ such implementations, of two types:

(1) Depthwise-separable 1D Convolution: We use two types of implementations (a) *Conv1D* refers to the default PyTorch (Paszke et al., 2019) kernel, that requires explicit padding (b) *Flash Conv1D* refers to fused kernel by FlashFFTConv (Fu et al., 2023b) - these are asymptotically quadratic in tile size $U$.

(2) FFT based convolution: we again use two implementations (a) PyTorch native, shown as *FFT*, that requires computing the FFTs of inputs, followed by pointwise multiplication and lastly performing the inverse FFT operation. (b) the second is provided by FlashFFTConv that does all the above in a single fused kernel shown as *FlashFFT*- both these scale as $O(U \log U)$.

## 4.3 MIXER ISOLATION STUDY AND HYBRIDIZATION OF $\tau$ IMPLEMENTATIONS

We evaluate the different convolution implementations for all settings of $B, D, M, U$ and observe that each of these four implementation lies on the pareto frontier curve of tile size vs latency as shown in Figure 3a. Our *Hybrid* approach dynamically chooses the best $\tau$ implementation for a given tile size $U$

based on the isolated empirically-measured efficiency of each implementation as shown in Figures 3a. Figure 3c shows the cumulative token time breakdown for the mixer and non-mixer components of the synthetic setup for all our $\tau$ implementations. We observe the non-mixer component staying largely the same across approaches - this is made possible through the employment of CUDA Graphs as further discussed in Appendix D.2.

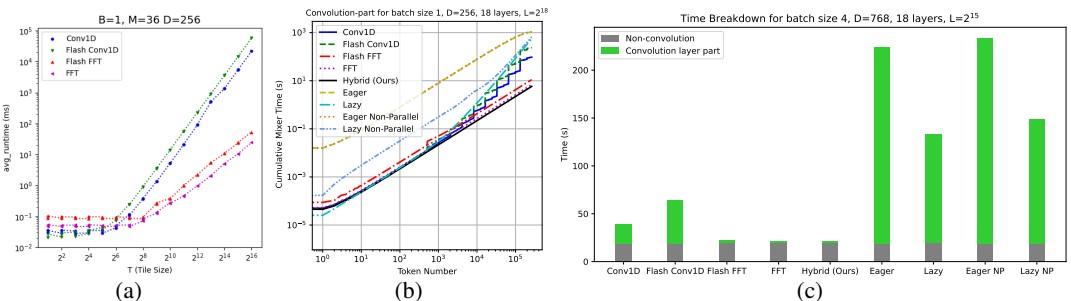

Figure 3: Mixer Isolation in a Synthetic setting: (a) Different implementations of $\tau$ are optimal for different tile sizes creating a pareto optimal curve for Hybrid to choose, (b) Cumulative mixer inference of Hybrid achieves the best of all $\tau$ Implementations (c) End-to-end cumulative token inference breakdown

## 4.4 IMPROVEMENTS JUSTIFICATION

This significant speed-up can be attributed to:

(1) The significantly lower $O(L \log^2 L)$ FLOPs required for our tile computation approach compared to the $\Omega(L^2)$ FLOPs required by *Eager* and *Lazy* counterparts. This is shown in Figure 2b where methods based on our tiling outperform by a large constant the quadratic methods in terms of time spent on the mixer components.

(2) Drastically reduced activation memory access from $\Omega(L^2)$ for *Eager* and *Lazy* to out $O(L \log L)$ tiling-based methods. This can be shown through the performance of *Flash Conv1D* (Figure 3b) which outperforms lazy and eager by a margin although it also performs $\Omega(L^2)$ FLOPs - it does so in a more memory-friendly and kernel-optimizable way.

(3) The dynamic choice of best $\tau$ implementation for given tile $U$ - hybrid outperforming any method using a fixed implementation (Figure 3b).

(4) Our engineering contributions: first, the DFT for the convolutional kernel is pre-computed ahead of time for all tile sizes. Second, Flash-FFT configurations are pre-initialized for these tile sizes to maximize hardware performance. Third, right padding is used instead of left padding to reduce computation time by half. Fourth, properties of circular convolution are exploited to halven FFT length. Finally, tile processing is parallelized across layers to saturate memory bandwidth for small tile computations and optimize computation for large tile computations, resulting in improved performance (one can notice improvements of $10-20\%$ even in the Eager and Lazy implementations).

## 5 CONCLUSION AND FURTHER WORK

We propose a framework for performing inference in certain autoregressive sequence models. Among such models, LCSMs are noteworthy: there, our framework provides an $O(L \log^2 L)$ inference algorithm which, when run empirically, yields up to $8\times$ end-to-end time-efficiency improvement. The framework exploits a causal, fractal tiling that helps save on data movement and share computation. Moreover, it allows for almost-complete across-layers parallelization of mixer-related workload.

An interesting future direction to pursue is that of designing architectures that fit out framework requirements and thus get fast-inference by construction. Furthermore, in the class of LCSMs, we have noted that one can achieve the same theoretical complexity if filters are made data-dependent, and, while previous works Arora et al. (2023); Karami & Ghodsi (2024) have shown the potential for these, they are not yet causal so looking into how to make filters data-dependent in a causal way is another promising direction.

## 6 ACKNOWLEDGMENTS

This work was partially enabled by use of the Kempner AI cluster. Sham Kakade acknowledges: this work has been made possible in part by a gift from the Chan Zuckerberg Initiative Foundation to establish the Kempner Institute for the Study of Natural and Artificial Intelligence. SK and CO also acknowledge support from the Office of Naval Research under award N00014-22-1-2377, and the National Science Foundation Grant under award #IIS 2229881.

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

## A    CONTEXTUAL DISCUSSION

In this section, we extend our discussion on the contextual relevancy of LCSMs and our method. We focus on the SISO (single-input-single-output) setting unless otherwise specified and assume that such a primitive is used across channels of the embedding dimensions to form a mixer.

As discussed in Section 2.3, LCSMs directly compute the filter $\rho$ to be used to map an input sequence $y \in \mathbb{R}^L$ to an output sequence $z \in \mathbb{R}^L$ via $z_t \triangleq \sum_{i=1}^t y_i \cdot \rho_{t-i}$. We assume here that $\rho \in \mathbb{R}^L$ although it could be longer since no entry past $L^{\text{th}}$ position matters. While $\rho$ is often underparameterized, one could, in principle, directly learn its entries (and, indeed, (Fu et al., 2023a) successfuly do that by using some extra regularization). However, even in underparameterized settings, where $\rho = f(\theta)$ for some $\theta \in \Theta$ with $\dim(\Theta) \ll L$, what filters $\rho$ we end up dealing with is

a function of data and $f$: here, $f$ implicitly bakes in a certain bias. While depending on bias (such as decaying pattern of $\rho$ as suggested by (Li et al., 2022)) one may assume $\rho$ to have certain exploitable structures, our method is broadly applicable regardless of such structural properties. That makes Flash Inference be bias and data-independent. For completeness, we mention in Section A.2 previous works exploiting such structure - that is, ways to perform fast inference assuming certain properties of $\rho$ hold.

Since, as mentioned in Section 2, there is a direct connection between long convolutions and LTI SSMs, we cover this more extensively in Section A.1.

## A.1 STATE SPACE MODELS

While SSMs and LCSMs can fundamentally represent the exact same class of models, they differ through their parameterization. Works such as (Poli et al., 2023; Li et al., 2022; Romero et al., 2021b;a; Shi et al., 2023; Fu et al., 2023a) try to characterize the convolution implicitly. On the other hand, LTI SSMs (Gu et al., 2021a; 2022; Ma et al., 2022; 2024; Gu et al., 2021b; Gupta et al., 2022; Mehta et al., 2022) define a recurrent system that is equivalent to a convolution and provide an alternative perspective in doing so. We define discrete SISO SSMs below following the notation of (Gu et al., 2021a).

State space models (SSM) can be thought of as linear RNNs. In its simplest form, SSM's mixer for a SISO operator mapping $y \in \mathbb{R}^L$ to $z \in \mathbb{R}^L$ is defined via an extra sequence of hidden states $u : \mathbb{R}^{L \times D'}$ where $D'$ is the SSM's dimension. For a fixed input $y : \mathbb{R}^L$, one defines:

$$u_t = A(t)u_{t-1} + B(t)y_t$$
$$z_t = C(t)^\top u_t$$

where $A(t) : \mathbb{R}^{D' \times D'}, B(t), C(t) : \mathbb{R}^{D'}$. While models like Mamba (Gu & Dao, 2023) let $A, B, C$ be functions of $y$ and, thus, vary with $t$, linear-time invariant (LTI) models assume $A$, $B$ and $C$ to be fixed parameters. Thus, the equations rewrite to:

$$u_t = Au_{t-1} + By_t$$
$$z_t = C^\top u_t$$

Opening up the recurrence, we get that:

$$z_t = \sum_{i=0}^{t} (C^\top A^{t-i} B) \cdot y_i$$

which corresponds to a convolution with the filter $\rho$ implicitly defined via $\rho_i = C^\top A^i B$. If $A$ is further assumed to be diagonal, we call the corresponding SSM diagonal. Gupta et al. (2022) showed that one does not lose practical model quality by assuming $A$ to be diagonal. When one tried to perform inference on an LTI SSMs, one could use:

- the recurrent view: keep around only the last state $u_{t-1}$ and update it in-place. This is why the memory of performing inference with an SSM scales with $D'$ rather than $L$. The time complexity of performing one step is $O(D'^2)$ in the general case, but as mentioned, one can resort to diagonal SSMs which drop this complexity to $O(D')$.
- convolutional view: precompute the equivalent convolution $\rho$ and use Flash Inference.

## A.2 COMPARISON TO STRUCTURAL FILTER EXPLOITATION

Following our discussion in Section A.1, LTI SSMs become a candidate of structural property that can be assumed on a filter $\rho \in \mathbb{R}^L$. While if one is to let $D'$ grow as much as $L$, it is provably possible to represent any filter $\rho$, it is a structural assumption to assume that a given filter $\rho$ can be represented (or well-approximated) by an LTI SSM of a particular dimension $D'$ (and this is what (Massaroli et al., 2024) do). However, if the filter has this structure then we can use the recurrent view and pay a time cost of $D'$ per token generation (assuming a diagonal SSM) while having our activation space drop by a factor of $L/D'$. Alternatively, we can store all the activations and use Flash Inference to get an average token step for $O(\log^2(L))$ time-budget. Hence, at least asymptotically,

if $D' \in O(\log^2(L))$, one would prefer the recurrent view (and thus using the $D'$-dimensional SSM equivalent to $\rho$). Past this point, our method is faster but still uses memory that scales up with $L$ rather than $D'$ so which one to choose is a function of our memory constraints.

Another noteworthy line of work follows dilated convolutional models and was popularized through (Shi et al., 2023), inspired in turn by (Van Den Oord et al., 2016). Paine et al. (2016) show how such models allow for linear time inference, while also needing to store all activations. Thus, granted such structure, Flash Inference is inferior and (Paine et al., 2016)'s approach should be used.

Finally, it is worth emphasizing that Flash Inference can accommodate data-dependent filters, while online exploitation of structure is often prohibitively expensive (as in the case of distilling $\rho$ into an LTI SSM following (Massaroli et al., 2024)). Moreover, the class of data-dependent convolution-based models is larger than that of LTI SSMs simply because of the linear-time invariant assumption: one cannot know the whole $A, B, C$ when only half of $\rho$ is unlocked (similarly to how data-dependent SSMs, such as (Gu & Dao, 2023), can represent more than LCSMs can). And while there are no major data-dependent causal LCSMs yet, works such as (Arora et al., 2023; Karami & Ghodsi, 2024) show the potential of such a research direction.

### A.3 ON FLASH INFERENCE'S PREFILLING AND AUTOREGRESSIVE GENERATION STAGES

We have touched briefly in Section 2.3.1 on how our method deals with prompts. In particular, if one is given a prompt of size $P$, FFT will be employed to fill in all the contributions of $y_{[1,P]}$ to $z_{[1,L]}$. The way this translates to the complete architecture is computing all of $a_{[1,P]}^{[0,M]}$ as well as the contributions of these to the future $L - P$ positions. One can do so at a fraction of the cost of the forward pass, by just caching the appropriate activations and then zeroing out positions in the range $[P + 1, L]$ before proceeding to the next layer (to not allow second order contributions to be performed). This comes at a cost that scales with $O(L \log L)$.

Afterwards, we can drop the first $P$ positions and apply our method starting with the current values of $a$ to perform autoregressive generation. This is because the eager nature of our prefill is more aggressive than in the case of transformers since we already account for all the relevant contributions of the prompt to the future. This can be very useful when we have a long prompt, relatively few autoregressive steps to perform and the autoregressive generation takes place on a different machine, since we save up valuable and communication costs.

We also want to emphasize, that Flash Inference focuses on optimizing the autoregressive generation part. However, if the prompt is large enough compared to the number of tokens generated, this may have smaller relative improvement since the overall time is dominated by the prefill.

## B THE FLASH INFERENCE FRAMEWORK

We propose a framework called Flash Inference to generalize our fast LCSM inference algorithm. To do so, we identify the main properties that are being exploited by Algorithm 2 and show how, granted these properties, one can get fast inference following the same kind of tiling.

Notice that in the case of LCSMs, it never mattered how one computes the contribution of a range of inputs to another range: only that it is well-defined (as soon as the inputs are available) and that it can be done more efficiently than the brute-force approach. Thus, the crux of Algorithm 2 is not convolution-specific, but rather the way we tile the space of contributions while ensuring autoregressive generation can take place.

### B.1 ARCHITECTURAL PROPERTIES

In order for our framework to apply, we need the mixers involved to have certain properties:

**P.1 Contribution-based** The used mixers work by aggregating contributions of each input position to each subsequent output position. That is, for any $\text{mixer}^\ell$, there exist an *associative* aggregation function $\text{agg} : \mathcal{X}^* \to \mathcal{X}$, a read function $\text{read} : \mathcal{X} \to \mathbb{R}^D$ and a contribution

function $cont : \mathbb{R}^D \times \mathbb{N} \times \mathbb{N} \to \mathcal{X}$ such that:

$$\text{mixer}(y)_i = \text{read}(\text{agg}(\text{cont}(y,1,i), \text{cont}(y,2,i), \ldots \text{cont}(y,i,i))) \tag{6}$$

where, $\mathcal{X}$ is a set of intermediate states and read is a function to map those back to embeddings. Recall that agg being associative means that $\text{agg}(x_1, x_2, \ldots x_\tau) = \text{agg}(\text{agg}(x_1, \ldots x_i), \text{agg}(x_{i+1}, \ldots x_\tau))$ for any $1 \leq i < \tau$. For a sensible architecture, the size of $\mathcal{X}$ and cost of agg should be of order $D$.

In the case of self-attention this translates to having read $\circ$ agg simulate the softmax, by letting $\mathcal{X} = \mathbb{R}^D \times \mathbb{R}$, $\text{agg} = \sum$ and

$$\text{cont}(y,i,j) = (V y_i \cdot e^{y_j^\top Q K^\top y_i}, e^{y_j^\top Q K^\top y_i}) = (v_i \cdot e^{\langle k_i, q_j \rangle}, e^{\langle k_i, q_j \rangle})$$

that is, the exponentially weighted value vector along with the exponential weight. Finally one can use $\text{read}(v, w) = v/w$ to implemnent the softmax normalization step.

In the case of LCSMs, one can simply choose $\mathcal{X} = \mathbb{R}^D$, read be the identity function, agg be the sum again and $\text{cont}(y,i,j) = y_i \odot \rho_{j-i}$.

**P.2** **Query-independent** The contribution function $\text{cont}(y,i,j)$ does not depend on $y_{[i+1,L]}$. Note that this is the case in LCSMs since $y_i \odot \rho_{j-i}$ only depends on $y_i$. However, it is not the case for transformers, since $\text{cont}(y,i,j)$ depends on $q_j$ which depends on $y_j$.

## B.2 SETTING AND DEFINITIONS

Suppose **P.1** holds and there exists an algorithm $\mathcal{A}$ that for any given input sequence $y$ and indices $l \leq r < l' \leq r'$, computes the contributions of $y_{[l,r]}$ to outputs at every position $p \in [l', r']$:

$$\mathcal{A}(y, [l,r], [l',r'])_{l' \leq p \leq r'} = \text{agg}(\text{cont}(y,l,p), \text{cont}(y,l+1,p), \ldots \text{cont}(y,r,p)).$$

Furthermore, for this choice of $\mathcal{A}$, let $\mathcal{T} : \mathbb{N} \times \mathbb{N} \to \mathbb{N}$ such that for any $l \leq r < l' \leq r'$, evaluating $A(y, [l,r], [l',r'])$ takes at most $\mathcal{T}(r - l + 1, r' - l' + 1)$ FLOPs.

If we dropped the $r < l'$ condition and set $l = l' = 1$ and $r = r' = L$, this algorithm would actually represent the procedure one needs to perform a forward pass during training - which we refer to as the *static* setting. In the case of LCSMs, $\mathcal{A}$ is the FFT-based algorithm underlying Lemma 1, with an associated $\mathcal{T}(L_1, L_2) = D(L_1 + L_2) \log(L_1 + L_2)$, as opposed to the naive implementation that would have $\mathcal{T}_{\text{naive}}(L_1, L_2) = D L_1 \cdot L_2$.

## B.3 MAIN RESULT

Our main result can be phrased as follows:

**Theorem 2.** *Under the assumptions **P.1** and **P.2**, one can generate $L = 2^P$ tokens autoregressively by performing, per layer, $L - 1$ black-box calls to $\mathcal{A}$ as well as $L$ more calls to* cont*, * agg*, * read *and* block*. Concretely, there are $L/2 = 2^{P-1}$ calls to $\mathcal{A}$ of length $1$ each, $L/4 = 2^{P-2}$ calls of length $2$ each and so on up to $1$ call of length $L/2$. Hence, neglecting the* cont*, * agg*, * read *and* block *part, the overall time complexity per mixer layer is:*

$$FLOPs = \sum_{q=0}^{P-1} 2^{P-1-q} \mathcal{T}(2^q, 2^q)$$

*Furthermore, during every token-generation iteration, the calls to $\mathcal{A}$ across different layers can be performed in parallel as there is no data-dependency between them.*

Here, we are being ignorant of the $L$ calls to cont, agg, read and block because they only scale linearly with $L$ - however MLPs scale quadratically in $D$ and therefore these can be the limiting factor when $D$ is large in comparison to $L$.

*Proof.* Figure 1 shows why the tiling used in Algorithm 2 covers every pair of contributions exactly once and in the correct order (remember, we only assumed agg to be associative, not necessarily commutative). The more general algorithm is illustrated in Algorithm 3 and follows the same shape as its LCSM counterpart. The only difference is that $\mathcal{X}$ is not assumed to be

$\mathbb{R}^D$ necessarily, so one cannot store the intermediate accumulated states in the same place they store the activations (though they could reuse some of the space since $a_i^\ell$ and $b_i^\ell$ never need to coexist). We use $b_i^\ell$ to store the inner part of Equation 6, namely $b_i^\ell$ incrementally computes $\mathrm{agg}(\mathrm{cont}(a^{\ell-1}, 1, i), \mathrm{cont}(a^{\ell-1}, 2, i), \dots \mathrm{cont}(a^{\ell-1}, i, i))$. Finally, we explicitly moved the gray tile calls to happen after the inner loop and in parallel - this simply shows the last point of the theorem, namely that the calls to $\mathcal{A}$ can be done parallelly across layers.

The only part remaining to be proved is the right counting of calls to $\mathcal{A}$ per layer: calls of length $2^q$ happen whenever $2^q$ divides $i$ but $2^{r+1}$ does not - that is, for $i \in \{2^q, 3 \cdot 2^q, 5 \cdot r^2, \dots (2^{P-q}-1) \cdot 2^q\}$, so $2^{P-q-1}$ calls. $\qquad\square$

---

**Algorithm 3** Generic Flash Inference

---

**Require:** Functions $\mathcal{A}, \mathrm{cont}, \mathrm{agg}, \mathrm{read}, \mathrm{block}^{[1,M]}$, first token $a_1^0$ and sampler
 1: **Output:** All activations $a^0, \dots, a^M \in \mathbb{R}^{L \times D}$ obtained by autoregressively sampling
 2: Initialize $b \in \mathcal{X}^{M \times L \times D}$ with elements neutral to $\mathrm{agg}$
 3: **for** $i \leftarrow 1$ **to** $L-1$ **do**
 4: $\quad U \leftarrow$ maximum power of 2 that divides $i$  # the side of the $i^{\text{th}}$ gray tile
 5: $\quad$ **for** $\ell \leftarrow 1$ **to** $M$ **do**
 6: $\quad\quad$ # account for the contribution of $a_i^{\ell-1}$ to $b_i^\ell$ - red cell
 7: $\quad\quad b_i^\ell = \mathrm{agg}(b_i^\ell, \mathrm{cont}(a^{\ell-1}, i, i))$
 8: $\quad\quad a_i^\ell = \mathrm{block}^\ell(\mathrm{read}(b_i^\ell))$
 9: $\quad$ **end for**
10: $\quad$ # account for the contribution of $a_{[i-U+1,i]}^{\cdot}$ to $b_{[i+1,i+U]}^{\cdot}$ - gray tile(s)
11: $\quad$ **parallelly across** $\ell \leftarrow 1$ **to** $M$ **do:**
12: $\quad\quad b_{[i+1,i+U]}^\ell = \mathrm{agg}(b_{[i+1,i+U]}^\ell, \mathcal{A}(a^{\ell-1}, [i-U+1,i], [i+1,i+U]))$
13: $\quad$ # generate next token based on the output of last layer at position $i$
14: $\quad a_{i+1}^0 = \mathrm{sampler}(a_i^M)$
15: **end for**

---

## C    Extension to Data-Dependent Filters

The reason why data-dependent filters are harder to deal with is that $\mathrm{cont}(y, i, j) = y_i \cdot \rho_{j-i}$ depends on both $y_i$ and on what $\rho_{j-i}$ depends on. By only assuming causality, we can only access $\rho_{j-i}$ after $z_{j-i-1}$ has been computed. This stops Algorithm 2 from working out-of-the-box since when $i$ is a power of 2, in order to account for the contribution of $a_{[1,i]}^{\ell-1}$ to $a_{[i+1,2i]}^\ell$ one will need access to $\rho_{[1,2i-1]}^\ell$ which is not guaranteed to be accessible (only $\rho_{[1,i]}^\ell$ can be assumed to be known at this stage).

The modified algorithm is shown in Algorithm 4 and precisely follows the tiling of van der Hoeven (1997) and, thus, its correctness transfers. Note that rather than using the $\mathcal{A}$ algorithm, it directly uses the untruncated convolution (implementable via FFT) - in the contribution space this corresponds to a parallelogram tile rather than rectangle - and this cannot be simulated via a rectangle, although the other way around is possible (and is what we did through Lemma 1). This implementation performs convolutions between two sequences of length $U$ each and uses the whole output and thus requires an order $2U$ FFT. Note that this happens twice for a given $i$ when $i+1$ is not a power of 2 (almost always). Algorithm 2, on the other hand, only performs a convolution between a length-$U$ sequence and a length-$2U$ sequence. However, as noted in Appendix D, we can get away without padding and thus performing only one order $2U$ FFT. Thus, our proposed tiling improves FLOPs by a factor of 2, but it requires the kernel to be data-independent.

## D    Implementation Improvements

In Algorithm 2, the calls to $\tau$ all have a shape of $\tau(y, [i-U+1, i], \rho, [i+1, i+U])$ where $U$ is a power of 2 that divides $i$. Following the proof of Lemma 1, we note that to implement $\tau$, we need to perform the convolution of a segment of length $U$ of $y$ and a prefix of length $2U$ of $\rho$. However,

---

**Algorithm 4** Flash Inference for LCSMs with data-dependent filters

---

**Require:** first token $a_0^1$, $a_0^\cdot$, $\rho_0^\cdot$, $\text{block}^{[1,M]}$ and sampler

1: **Output:** All activations $a^0, \ldots, a^M \in \mathbb{R}^{L \times D}$ obtained by autoregressively sampling
2: Initialize $a$ with zeros outside $a_0^\cdot$
3: **for** $i \leftarrow 1$ **to** $L - 1$ **do**
4:   $U \leftarrow$ maximum power of 2 that divides $(i + 1)$
5:   **for** $\ell \leftarrow 1$ **to** $M$ **do**
6:    Compute $\rho_i^\ell$ as a causal function of data $a_{[1,i]}^{\ell-1}$ as per model specification
7:    # account for the newly available contributions
8:    $a_i^\ell = \text{block}^\ell(a_i^\ell + a_i^{\ell-1} * \rho_0^\ell + a_0^{\ell-1} \odot \rho_i^\ell)$
9:    # account for some eager contributions
10:    **if** $i + 1 = U$ **then**
11:     # downgrade the power of 2 by half if $i + 1$ is already a power of 2
12:     $U \leftarrow U/2$
13:     $a_{[2U,4U-2]}^\ell \mathrel{+}= CONV(a_{[U,2U-1]}^{\ell-1}, \rho_{[U,2U-1]}^\ell)$
14:    **else**
15:     $a_{[i+1,i+2U-1]}^\ell \mathrel{+}= CONV(a_{[U,2U-1]}^{\ell-1}, \rho_{[i-U+1,i]}^\ell)$
16:     $a_{[i+1,i+2U-1]}^\ell \mathrel{+}= CONV(\rho_{[U,2U-1]}^\ell, a_{[i-U+1,i]}^{\ell-1})$
17:    **end if**
18:   **end for**
19:   # generate next token based on the output of last layer at position $i$
20:   $a_{i+1}^0 = \text{sampler}(a_i^M)$
21: **end for**

---

following the convolution, of the $3U - 1$ outputs, indexed $[0, 3U - 2]$, we are only interested in the middle $U$ of them, namely indices $[U, 2U - 1]$. The canonical way of performing convolutions via FFT involves padding by enough $0s$ and using an order large enough to store the whole output which rounded up to the closest power of 2 would mean an order $4U$ FFT. However, using a $2U$ FFT call, which will perform a cyclical convolution, is enough for our purposes, since the values of interest are not affected by the cyclicity - that is, when folding outputs at $[2U, 3U - 2]$ onto $[0, U - 2]$, we do not intersect $[U, 2U - 1]$. Furthermore, there are only $\log L$ different lengths of prefixes of $\rho$ involved in these convolution so one could, in fact, cache the DFTs for these as a precomputation step, thus dropping the number of DFT's per convolution from 3 to 2, speeding up by a further $\times 1.5$ factor.

### D.1 MORE IMPLEMENTATION DETAILS

In Section 4.2, it has been mentioned that we considered 7 implementations of $\tau$. Besides the four mentioned (Torch's and FlashFFT's Conv1D and FFT) we also did 3 more naive versions:

1. Torch FFT (no precomputed FFT for the convolution kernel): This implementation is identical to Torch-FFT except that we do not pre-compute the DFT of the convolution kernel beforehand.

2. Vectorized Conv1D: It implements a 1D convolution operation in a vectorized manner using tensors and leverages efficient operations to process multiple slices in parallel. It takes advantage of the fact that convolutions are depth-wise separable.

3. Element-wise Conv1D: This implementation does an element-wise dot product of the convolution kernel with the input, while iteratively sliding the kernel over the input.

### D.2 CPU OVERHEAD: THE SILENT PERFORMANCE KILLER

One primary improvement we introduce is the use of CUDA Graphs to mitigate the CPU overhead observed in the non-mixer components during inference.

In both synthetic settings and the Hyena model, these non-mixer components primarily deal with only current token being generated. As a result, the GPU compute time is often shorter than the kernel dispatch time, leading to CPU overhead. Typically, this overhead remains hidden when the

mixer component requires significant GPU compute time. However, Flash Inference dramatically reduces mixer times (up to $120\times$), exposing the CPU overhead and causing the observed decrease in end-to-end speed-ups. A similar effect is observed for the Lazy approach compared to Eager, as Lazy's faster execution also exposes the overhead.

To address this, we employ CUDA Graphs, which record all kernel dispatches for generating a single token into a graph. For subsequent token generation, the same graph is replayed, allowing all kernels for the non-mixer components to be dispatched simultaneously. This approach significantly reduces CPU overhead and ensures consistent overhead times for both Flash Inference and its baselines.

With these improvements, Flash Inference demonstrates much better end-to-end relative gains, as the mixer now constitutes a larger portion of the total runtime. We observe speed-ups of up to $120\times$ in the mixer and up to $8\times$ in the end-to-end Hyena runs (as opposed to $1.6\times$ when not using CUDA Graphs). All the results are shown in Appendix I.

## E    STORING ONLY HALF OF THE ACTIVATIONS

If the whole activation tensor ($M \times L \times D$) cannot fit the memory of one GPU, we can save a factor of 2 along the $L$-axis. To do this, observe that after $(L/2)^{\text{th}}$ iteration completes we never need to look back to any activation at position $i \leq L/2$. Hence, one can hope to reuse the space used for the first half to store the second half of activations. While doing this directly by having $\mathcal{A}$ work as an in-place operator requires storing $MLD$ intermediate values, one can choose to not apply $\mathcal{A}$ parallely across all layers (or even dimensions), as discussed in Section 3.3. Processing the tiles sequentially allows for the reuse of the space for interim results; once the result of $\mathcal{A}$ is read, we place it back to where the input used to be - similarly in nature to gradient accumulation - always placing the output back into the input slot. This way, the peak extra memory required for the call to $\mathcal{A}$ can be $LD$ or even $L$.

As we get the output of $\mathcal{A}(a^0, 1, L/2, L/2 + 1, L)$, which has a shape of $(L/2) \times D$, we overwrite $a^0$ by it and then move on to the next layer. We can do this because the contribution of $a_i^0$ to $a_i^1$ has already been properly accounted for so the first half of $a^0$ truly becomes irrelevant for next layers and iterations to come. Hence, $a^0$ contains the intended second half of $a^1$. We then proceed to do the same thing for $a^1$ and so on. At the end, $a^{[0\ldots M-1]}$ will contain the second halves of $a^{[1\ldots M]}$, so we can shift these by 1, emptying $a^0$ and now finally having $a$ represent the second halves of activations - the first halves have been discarded.

Although per layer, one will have a peak memory containing both $a^\ell$, the output, and whatever other necessary temporary values needed to perform $\mathcal{A}$, this does not scale with $M$ - because we essentially reuse the extra space needed by $\mathcal{A}$ across different layers. We seemingly pay the cost of not performing the calls to $\mathcal{A}$ in parallel across layers, but if storing the $M \times L \times D$ tensor of activations was an issue, then one would very likely have to make this sort of compromise to be able to run $\mathcal{A}$ in the first place.

## F    DISCUSSION ON MEMORY

Besides peak memory usage, it is important to understand how memory boundedness affects end-to-end performance. In general, GPUs have 2 types of memory bound:

- Memory Latency-Bound: most time is spent on memory operations, but parallelism is insufficient to saturate GPU memory bandwidth. In this setting, throughput can be optimized by increasing parallelism to saturate memory bandwidth.

- Memory Bandwidth-Bound: GPU memory bandwidth is already saturated with available parallelism. In this case, throughput optimization is achieved by increasing arithmetic intensity (compute-to-IO ratio), usually by changing the algorithm or fusing operations.

Putting this into perspective for our use cases, and looking at how across-layer parallelization affects performance, we have that:

- Lazy computes token contributions using pointwise multiplication kernels, accessing almost all sequence activations for large $i$ values, making it memory bandwidth-bound beyond a

certain (early) iteration index. Hence across-layer parallelization helps most in the beginning. Lazy acts exactly as a dual (it is memory bandwidth-bound until close to the end).

- Our approach computes token contributions in a tiled fashion, with most tiles being small (87.5% have size $\leq 4$), making the workload memory-latency bound. Parallelizing across layers can then further saturate memory bandwidth. For large tile sizes we end up using FFT implementations to decrease arithmetic intensity since we get to a compute-bound regime.

In Section 3.3, we discussed the peak memory usage being given by the gray calls to $\tau$. Since each call to $\tau$ performs $D$ FFTs of length $L$ each (when dealing with the biggest tile of side $L/2$), the naive implementation would use up to $MD \cdot L$ extra memory to perform these FFT calls. However, this assumes the biggest gray tile is being processed in parallel across all layers and across all dimensions at once. We do not need to maximize parallelization on all sizes of tiles: for the largest ones, we can process the tiles at different layers sequentially by reusing the space, thus dropping to an overhead of $O(LD)$, or even just $O(L)$ if one is to treat sequentially the different dimensions. Dropping parallelization could theoretically yield a slowdown, but that is only when we are not memory bandwidth-bound which is the case if one has to worry about allocating more than $O(LD)$ - in that case, the FFT calls are anyway not happening in parallel. Hence, one can easily drop the peak memory overhead to $O(LD)$ or even $O(L)$ without incurring an actual time cost.

# G    FRAMEWORK ADAPTATIONS

We covered the clean case of LCSMs and of other architectures that fit Theorem 2. However, all these assume complete causality of the architecture as well as all the layers fitting our framework. This need not be the case and Flash Inference can easily be adapted to deal with both.

As far as causality is concerned, one needs to assume it for autoregressive inference to be well-defined. Remember, we defined causality by having $a^\ell(x)_i$ be a function of only $x_{[1,i]}$ or, equivalently, $\text{mixer}^\ell(y)_i$ is a function of $y_{[1..i]}$. If this was not the case, then we could not compute $x_{i+1}$ by only knowing $x_i$. However, much like the encoder-decoder architecture, we can allow for non-causal prompts. That is, if at any point an external user gives us a sequence of tokens (say an image), we could fill in the activations at those position in the standard forward-pass way, and then start our procedure from there as if we were given a fixed a prompt. Granted a prompt and the activations at its respective positions, we can proceed with our inference method as described in Section 2.3.1, following (Massaroli et al., 2024, Lemma 2.1), and essentially accounting in advance for the contribution of all this prompt to all subsequent positions.

Another useful adaption for most alternatives to attention is that of using hybrids: architectures where different layers use different mixers. Our method can be applied seamlessly to the subset of layers that fit the framework. All the algorithm stays unchanged but computing $a^\ell(x)_i$ for the layers $\ell$ which do not fit our framework needs to be done in the way specific to $\text{mixer}^\ell$. Importantly, even the across-layer parallelization can be maintained since the gray tiles only affect subsequent positions at the layers that fit the framework, hence not interfering with the rest. At any point in our algorithm, we have computed every piece of information that the lazy approach has, as well as some extra partial future contribution. Therefore, everything that worked assuming the lazy approach will work with our method as well - this includes other layers attending to more than the previous layer.

# H    PROOFS

**Lemma.** Let $1 \leq l \leq r \leq l' \leq r' \leq L$ represent ranges of lengths $L_1 = r - l + 1$ and $L_2 = r' - l' + 1$ of $y$ and $z$, respectively. There exists an FFT-based algorithm running in $O(L_1 + L_2)$ space and $O((L_1 + L_2) \log(L_1 + L_2))$ time complexity that, given access to $y_{[l,r]}$, computes all the elements of $\tau(y, [l, r], \rho, [l', r'])_{[l', r']}$ - that is, all the aggregated contributions of $y_{[l,r]}$ to all of $z_{[l', r']}$.

*Proof.* Consider performing a convolution between $y_{[l,r]}$ and $\rho_{[l'-r,r'-l]}$. It holds that for each $j \in [l', r']$ and $i \in [l, r]$, $j - i$ is in the range of $\rho$. On the other hand, truncating the output of the convolution appropriately, one can keep only the corresponding outputs for each $j \in [l', r']$. The overall size of the FFT is then given by the sum of the lengths, which is at most $r - l + 1 + (r' - $

$l) - (l' - r) + 1 = 2(r - l + 1) + (r' - l' + 1) - 1 = O((r - l + 1) + (r' - l' + 1))$. Since an FFT of order $L$ runs in $O(L \log L)$ and takes up $O(L)$ memory, the conclusion follows. $\qquad\square$

# I EXTENDED EXPERIMENTAL RESULTS AND ANALYSIS

## I.1 BAR PLOTS

For completeness, we provide the breakdown for both Hyena(Figure 4) and synthetic(Figure 5) settings. As mentioned in the main body, we see consistency across the non-convolution part and very significant improvement on the convolution part.

## I.2 SUMMARY

We provide the end-to-end relative speedups for our methods with regards to Lazy (the parallel version) for all the 21 runs that we performed, for Hyena as well as the synthetic setup, measuring both complete wall-clock time, as well as mixer time.

Notice that in some of the synthetic runs, the torch-based FFT implementation is faster than Hybrid - we attribute that to noise since the absolute runtime difference is below 50 ms and it translates to less than 2% drop in performance. Hybrid however sometimes exceeds FFT's performance by close to 20% (for example in the very first experiment).

Worth noticing is also the consistent 10% speedup that the parallelization of Lazy yielded - do note that the reported speedups are relative to the parallel version of Lazy, but they are even bigger relative to the non-parallel version. Furthermore, Conv1D[1], yields a consistent $> 5$-fold improvement while still having a quadratic complexity - this shows the potential of our framework as a grouping mechanism even when no asymptotically faster algorithms to deal with the tiles exist: we simply get better arithmetic intensity.

---

[1]The Conv1D kernel crashes for configurations $B = 1, N = [18, 36], L = [2^{17}, 2^{16}]$. This is because it requires 32-bit indexing which is only supported in *cuDNN* version 9.3 and above. Our experiments use *cuDNN* v9.1.

Table 1: Mixer times improvements on end-to-end Hyena relative to the lazy approach (higher is better)

| B | N | D | L | Hybrid | Eager | Eager_NP | Lazy_NP |
|---|---|---|---|--------|-------|----------|---------|
| 1 | 18 | 256 | $2^{18}$ | 95.96 $\times$ | 0.54 $\times$ | 0.52 $\times$ | 0.85 $\times$ |
|   |    | 768 | $2^{17}$ | 100.38 $\times$ | 0.54 $\times$ | 0.53 $\times$ | 0.89 $\times$ |
|   |    | 864 | $2^{17}$ | 110.74 $\times$ | 0.53 $\times$ | 0.53 $\times$ | 0.91 $\times$ |
|   | 36 | 256 | $2^{17}$ | 79.71 $\times$ | 0.54 $\times$ | 0.50 $\times$ | 0.76 $\times$ |
|   |    | 768 | $2^{16}$ | 89.36 $\times$ | 0.54 $\times$ | 0.51 $\times$ | 0.82 $\times$ |
|   |    | 864 | $2^{16}$ | 94.11 $\times$ | 0.54 $\times$ | 0.52 $\times$ | 0.84 $\times$ |
| 2 | 18 | 256 | $2^{17}$ | 75.22 $\times$ | 0.54 $\times$ | 0.50 $\times$ | 0.85 $\times$ |
|   |    | 768 | $2^{16}$ | 84.88 $\times$ | 0.54 $\times$ | 0.51 $\times$ | 0.87 $\times$ |
|   |    | 864 | $2^{16}$ | 93.35 $\times$ | 0.54 $\times$ | 0.51 $\times$ | 0.86 $\times$ |
|   | 36 | 256 | $2^{16}$ | 64.60 $\times$ | 0.54 $\times$ | 0.49 $\times$ | 0.77 $\times$ |
|   |    | 768 | $2^{15}$ | 74.80 $\times$ | 0.55 $\times$ | 0.50 $\times$ | 0.80 $\times$ |
|   |    | 864 | $2^{15}$ | 81.20 $\times$ | 0.55 $\times$ | 0.49 $\times$ | 0.77 $\times$ |
| 4 | 18 | 256 | $2^{16}$ | 66.82 $\times$ | 0.54 $\times$ | 0.53 $\times$ | 0.88 $\times$ |
|   |    | 768 | $2^{15}$ | 78.79 $\times$ | 0.55 $\times$ | 0.53 $\times$ | 0.87 $\times$ |
|   |    | 864 | $2^{15}$ | 84.77 $\times$ | 0.55 $\times$ | 0.52 $\times$ | 0.88 $\times$ |
|   | 36 | 256 | $2^{16}$ | 74.92 $\times$ | 0.54 $\times$ | 0.49 $\times$ | 0.79 $\times$ |
|   |    | 768 | $2^{15}$ | 88.22 $\times$ | 0.56 $\times$ | 0.51 $\times$ | 0.84 $\times$ |
|   |    | 864 | $2^{15}$ | 92.96 $\times$ | 0.56 $\times$ | 0.49 $\times$ | 0.81 $\times$ |
| 8 | 18 | 256 | $2^{16}$ | 81.37 $\times$ | 0.55 $\times$ | 0.53 $\times$ | 0.90 $\times$ |
|   |    | 768 | $2^{15}$ | 96.14 $\times$ | 0.56 $\times$ | 0.53 $\times$ | 0.90 $\times$ |
|   |    | 864 | $2^{15}$ | 104.81 $\times$ | 0.56 $\times$ | 0.53 $\times$ | 0.91 $\times$ |

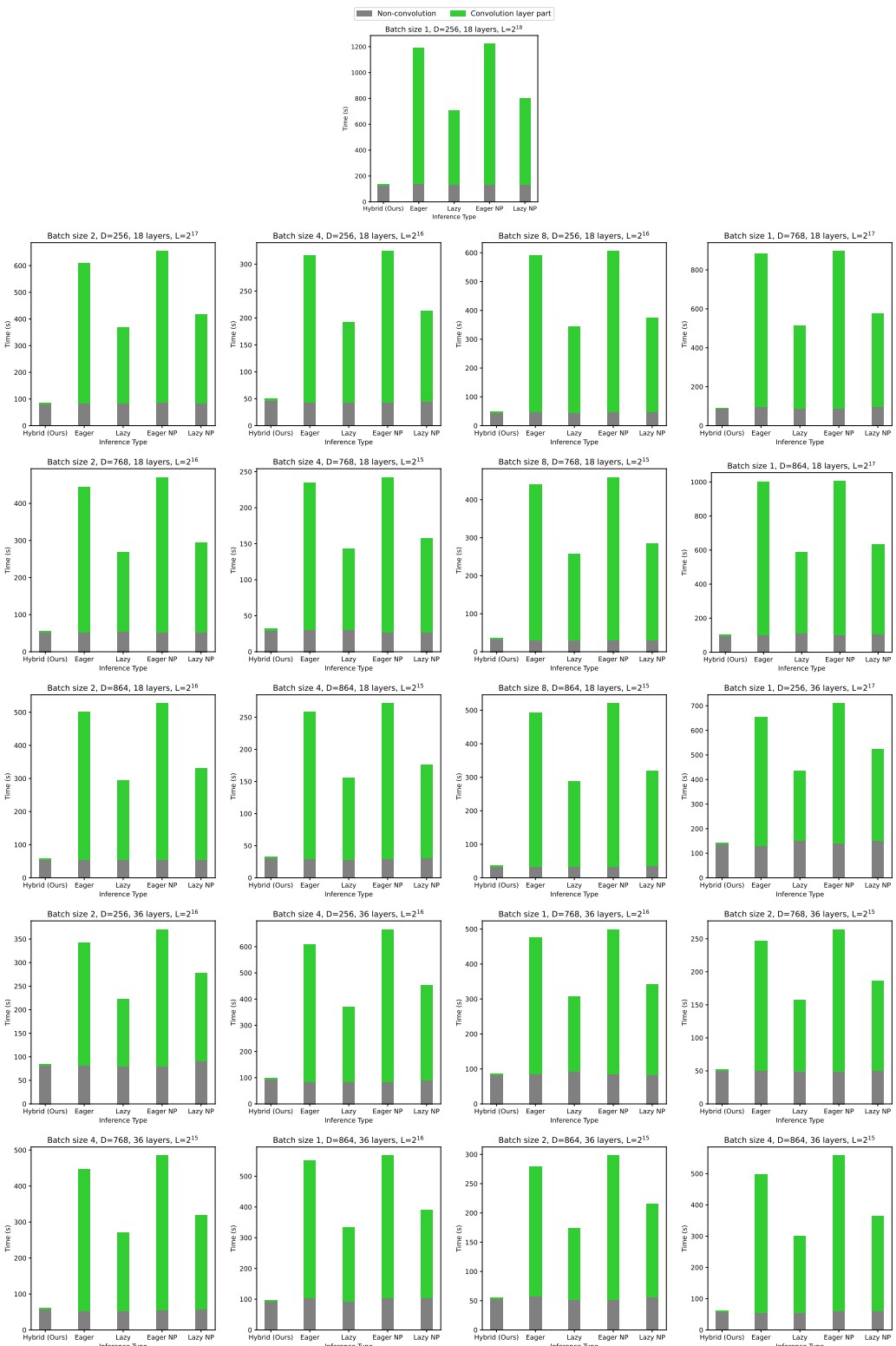

Figure 4: Time breakdown for end-to-end Hyena experiments.

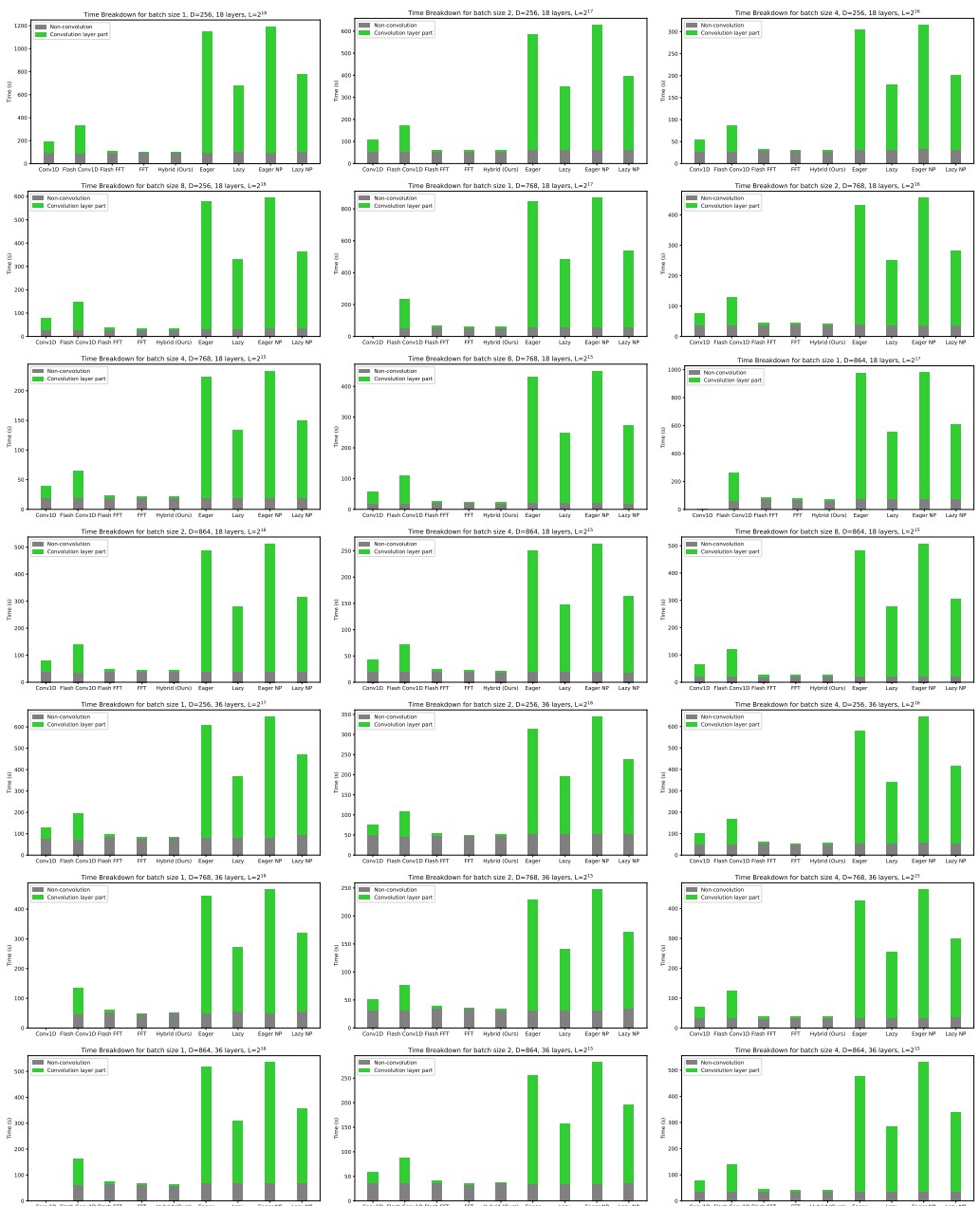

Figure 5: Time breakdown for end-to-end synthetic experiments.

Table 2: Cummulative times improvements on end-to-end Hyena relative to the lazy approach (higher is better)

| B | N | D | L | Hybrid | Eager | Eager_NP | Lazy_NP |
|---|---|---|---|---|---|---|---|
| 1 | 18 | 256 | $2^{18}$ | $5.30 \times$ | $0.59 \times$ | $0.57 \times$ | $0.88 \times$ |
| | | 768 | $2^{17}$ | $5.60 \times$ | $0.58 \times$ | $0.57 \times$ | $0.89 \times$ |
| | | 864 | $2^{17}$ | $5.71 \times$ | $0.59 \times$ | $0.59 \times$ | $0.93 \times$ |
| | 36 | 256 | $2^{17}$ | $3.11 \times$ | $0.66 \times$ | $0.61 \times$ | $0.83 \times$ |
| | | 768 | $2^{16}$ | $3.60 \times$ | $0.64 \times$ | $0.62 \times$ | $0.90 \times$ |
| | | 864 | $2^{16}$ | $3.50 \times$ | $0.60 \times$ | $0.59 \times$ | $0.85 \times$ |
| 2 | 18 | 256 | $2^{17}$ | $4.30 \times$ | $0.61 \times$ | $0.56 \times$ | $0.88 \times$ |
| | | 768 | $2^{16}$ | $4.81 \times$ | $0.60 \times$ | $0.57 \times$ | $0.91 \times$ |
| | | 864 | $2^{16}$ | $5.09 \times$ | $0.59 \times$ | $0.56 \times$ | $0.89 \times$ |
| | 36 | 256 | $2^{16}$ | $2.63 \times$ | $0.65 \times$ | $0.60 \times$ | $0.80 \times$ |
| | | 768 | $2^{15}$ | $3.01 \times$ | $0.64 \times$ | $0.60 \times$ | $0.85 \times$ |
| | | 864 | $2^{15}$ | $3.16 \times$ | $0.62 \times$ | $0.58 \times$ | $0.80 \times$ |
| 4 | 18 | 256 | $2^{16}$ | $3.88 \times$ | $0.61 \times$ | $0.59 \times$ | $0.90 \times$ |
| | | 768 | $2^{15}$ | $4.53 \times$ | $0.61 \times$ | $0.59 \times$ | $0.91 \times$ |
| | | 864 | $2^{15}$ | $4.85 \times$ | $0.60 \times$ | $0.57 \times$ | $0.89 \times$ |
| | 36 | 256 | $2^{16}$ | $3.76 \times$ | $0.61 \times$ | $0.55 \times$ | $0.81 \times$ |
| | | 768 | $2^{15}$ | $4.46 \times$ | $0.61 \times$ | $0.56 \times$ | $0.85 \times$ |
| | | 864 | $2^{15}$ | $4.86 \times$ | $0.61 \times$ | $0.54 \times$ | $0.82 \times$ |
| 8 | 18 | 256 | $2^{16}$ | $6.95 \times$ | $0.58 \times$ | $0.57 \times$ | $0.92 \times$ |
| | | 768 | $2^{15}$ | $7.14 \times$ | $0.59 \times$ | $0.56 \times$ | $0.91 \times$ |
| | | 864 | $2^{15}$ | $7.83 \times$ | $0.59 \times$ | $0.56 \times$ | $0.91 \times$ |

Table 3: Mixer times improvements on end-to-end synthetic relative to the lazy approach (higher is better)

| B | N | D | L | Hybrid | C1D | flashC1D | flashFFT | FFT | Eager | Eager_NP | Lazy_NP |
|---|---|---|---|--------|-----|----------|----------|-----|-------|----------|---------|
| 1 | 18 | 256 | $2^{18}$ | 97.70 × | 5.94 × | 2.40 × | 51.16 × | 82.27 × | 0.54 × | 0.52 × | 0.85 × |
|   |    | 768 | $2^{17}$ | 112.62 × | - | 2.42 × | 51.18 × | 111.57 × | 0.54 × | 0.53 × | 0.90 × |
|   |    | 864 | $2^{17}$ | 124.30 × | - | 2.42 × | 56.14 × | 119.41 × | 0.53 × | 0.53 × | 0.91 × |
|   | 36 | 256 | $2^{17}$ | 82.69 × | 5.94 × | 2.41 × | 39.73 × | 78.58 × | 0.54 × | 0.50 × | 0.76 × |
|   |    | 768 | $2^{16}$ | 100.53 × | - | 2.41 × | 35.36 × | 102.32 × | 0.54 × | 0.51 × | 0.82 × |
|   |    | 864 | $2^{16}$ | 112.33 × | - | 2.41 × | 36.58 × | 112.81 × | 0.54 × | 0.52 × | 0.84 × |
| 2 | 18 | 256 | $2^{17}$ | 84.84 × | 5.94 × | 2.41 × | 52.28 × | 81.43 × | 0.54 × | 0.50 × | 0.85 × |
|   |    | 768 | $2^{16}$ | 84.84 × | 5.81 × | 2.40 × | 45.94 × | 88.36 × | 0.54 × | 0.51 × | 0.87 × |
|   |    | 864 | $2^{16}$ | 97.23 × | 5.85 × | 2.40 × | 49.77 × | 94.40 × | 0.54 × | 0.51 × | 0.87 × |
|   | 36 | 256 | $2^{16}$ | 69.28 × | 5.74 × | 2.40 × | 39.57 × | 69.03 × | 0.54 × | 0.49 × | 0.77 × |
|   |    | 768 | $2^{15}$ | 78.79 × | 5.54 × | 2.43 × | 31.91 × | 76.47 × | 0.55 × | 0.51 × | 0.80 × |
|   |    | 864 | $2^{15}$ | 85.89 × | 5.56 × | 2.43 × | 33.80 × | 87.56 × | 0.55 × | 0.49 × | 0.77 × |
| 4 | 18 | 256 | $2^{16}$ | 75.13 × | 5.97 × | 2.49 × | 47.70 × | 74.52 × | 0.54 × | 0.53 × | 0.88 × |
|   |    | 768 | $2^{15}$ | 80.48 × | 5.75 × | 2.52 × | 42.66 × | 82.05 × | 0.55 × | 0.53 × | 0.87 × |
|   |    | 864 | $2^{15}$ | 93.40 × | 5.79 × | 2.54 × | 45.22 × | 88.84 × | 0.55 × | 0.52 × | 0.88 × |
|   | 36 | 256 | $2^{16}$ | 79.85 × | 5.78 × | 2.40 × | 50.57 × | 80.60 × | 0.54 × | 0.49 × | 0.79 × |
|   |    | 768 | $2^{15}$ | 92.30 × | 6.47 × | 2.45 × | 46.37 × | 88.90 × | 0.56 × | 0.51 × | 0.84 × |
|   |    | 864 | $2^{15}$ | 96.29 × | 6.49 × | 2.45 × | 47.93 × | 94.17 × | 0.56 × | 0.50 × | 0.81 × |
| 8 | 18 | 256 | $2^{16}$ | 95.32 × | 6.10 × | 2.51 × | 56.11 × | 87.85 × | 0.55 × | 0.53 × | 0.90 × |
|   |    | 768 | $2^{15}$ | 102.49 × | 5.89 × | 2.55 × | 54.83 × | 94.08 × | 0.56 × | 0.53 × | 0.90 × |
|   |    | 864 | $2^{15}$ | 107.36 × | 5.93 × | 2.55 × | 45.75 × | 98.61 × | 0.56 × | 0.53 × | 0.91 × |

Table 4: Cummulative times improvements on end-to-end synthetic relative to the lazy approach (higher is better)

| B | N | D | L | Hybrid | C1D | flashC1D | flashFFT | FFT | Eager | Eager_NP | Lazy_NP |
|---|---|---|---|--------|-----|----------|----------|-----|-------|----------|---------|
| 1 | 18 | 256 | $2^{18}$ | 6.70 × | 3.55 × | 2.04 × | 6.34 × | 6.64 × | 0.59 × | 0.57 × | 0.87 × |
| | | 768 | $2^{17}$ | 8.36 × | - | 2.09 × | 7.08 × | 8.05 × | 0.57 × | 0.56 × | 0.91 × |
| | | 864 | $2^{17}$ | 7.76 × | - | 2.09 × | 6.66 × | 7.42 × | 0.57 × | 0.56 × | 0.91 × |
| | 36 | 256 | $2^{17}$ | 4.32 × | 2.91 × | 1.89 × | 3.88 × | 4.48 × | 0.60 × | 0.57 × | 0.78 × |
| | | 768 | $2^{16}$ | 5.17 × | - | 2.00 × | 4.56 × | 5.50 × | 0.61 × | 0.58 × | 0.85 × |
| | | 864 | $2^{16}$ | 5.01 × | - | 1.91 × | 4.30 × | 4.78 × | 0.60 × | 0.58 × | 0.87 × |
| 2 | 18 | 256 | $2^{17}$ | 6.02 × | 3.30 × | 2.01 × | 5.70 × | 6.01 × | 0.59 × | 0.55 × | 0.88 × |
| | | 768 | $2^{16}$ | 6.06 × | 3.34 × | 1.97 × | 5.75 × | 5.90 × | 0.58 × | 0.55 × | 0.89 × |
| | | 864 | $2^{16}$ | 6.38 × | 3.50 × | 2.02 × | 6.03 × | 6.37 × | 0.57 × | 0.54 × | 0.88 × |
| | 36 | 256 | $2^{16}$ | 3.84 × | 2.59 × | 1.84 × | 3.67 × | 3.94 × | 0.63 × | 0.57 × | 0.82 × |
| | | 768 | $2^{15}$ | 4.25 × | 2.74 × | 1.84 × | 3.69 × | 3.89 × | 0.62 × | 0.57 × | 0.83 × |
| | | 864 | $2^{15}$ | 4.19 × | 2.71 × | 1.83 × | 3.88 × | 4.51 × | 0.62 × | 0.56 × | 0.81 × |
| 4 | 18 | 256 | $2^{16}$ | 5.83 × | 3.34 × | 2.07 × | 5.46 × | 5.77 × | 0.59 × | 0.57 × | 0.89 × |
| | | 768 | $2^{15}$ | 6.25 × | 3.45 × | 2.08 × | 5.91 × | 6.29 × | 0.60 × | 0.57 × | 0.89 × |
| | | 864 | $2^{15}$ | 7.15 × | 3.45 × | 2.08 × | 6.24 × | 6.64 × | 0.59 × | 0.56 × | 0.90 × |
| | 36 | 256 | $2^{16}$ | 5.98 × | 3.33 × | 2.02 × | 5.49 × | 6.38 × | 0.59 × | 0.53 × | 0.82 × |
| | | 768 | $2^{15}$ | 6.62 × | 3.64 × | 2.06 × | 6.75 × | 6.59 × | 0.59 × | 0.55 × | 0.85 × |
| | | 864 | $2^{15}$ | 6.99 × | 3.74 × | 2.06 × | 6.68 × | 7.13 × | 0.59 × | 0.53 × | 0.84 × |
| 8 | 18 | 256 | $2^{16}$ | 9.88 × | 4.19 × | 2.22 × | 9.14 × | 10.36 × | 0.57 × | 0.56 × | 0.90 × |
| | | 768 | $2^{15}$ | 11.55 × | 4.30 × | 2.29 × | 9.85 × | 10.55 × | 0.58 × | 0.55 × | 0.91 × |
| | | 864 | $2^{15}$ | 11.37 × | 4.25 × | 2.28 × | 10.56 × | 11.20 × | 0.57 × | 0.55 × | 0.91 × |

