# OpenReview forum: "Flash Inference: Near Linear Time Inference for Long Convolution Sequence Models and Beyond"
_ICLR.cc/2025/Conference — ICLR 2025 Poster_

### Official Review · Reviewer_byxw · 2024-11-03

**Soundness:** 3
**Presentation:** 3
**Contribution:** 2
**Rating:** 6
**Confidence:** 2

**Summary:**

This paper develops a computation framework for efficient inference of long convolution sequence models (LCSMs). The proposed approach is based on relaxed polynomial interpolation, and can speed up LCSMs’ exact inference to the quasilinear time, as is shown by both complexity analysis and numerical verifications.

**Strengths:**

1. The paper is well-written and easy to follow.
2. The complexity analysis regarding both the computation and memory is detailed.
3. The experiments are also provided to numerically verify the effectiveness of the proposed method.

**Weaknesses:**

1. The main concern is about the novelty of this paper, particularly compared to the work by van der Hoeven and "dynamic FFT". It would be clearer if authors can provide more details about the differences or improvements of this work compared to former references.
2. The applied setting seems somewhat limited, since it currently works only for Hyena-related architectures. Does the proposed method have the potential to inspire further extensions for general state-space models (SSMs; as with an equivalent convolutional filter) and Transformers (despite reasons stated in Sec. 3.4)?
3. The numerical experiments part can be strengthened by e.g. adding tests for multiple configurations of hyper-parameters.

**Questions:**

1. Please kindly provide more details to the questions raised in the "Weaknesses" section.
2. How about the case when the non-mixers’ runtime is dominant, e.g. the large MLP module is common in practice (such as Transformers in applications). Do the improvements shown in Fig. 2(a) & 3(c) become marginal?
3. For Fig. 3(c), why is the non-mixers’ runtime less for Eager (NP) & Lazy (NP)? In addition, it seems that there is no reduction of mixers’ runtime of Hybrid compared to (Flash) FFT?

**Details Of Ethics Concerns:**

There are no ethics concerns.

---

> ### Author Response · Authors · 2024-11-23
> **First revision rebuttal**
>
> Thank you for the comments! We tried to address them in the first revision of the paper and below:
>
> Regarding weaknesses:
> - We use a different tiling from that of van der Hoeven to exploit data-independent filters. We have a significant number of systems improvements to make the method practical, that we enumerate and emphasize in several parts in the paper (in the intro, as well as the improvements justification section). Among these, the across-layer parallelization stands out as an optimization that is specific to such layered architectures. We also show how to halven the memory usage, how to exploit DFT precomputations, dynamically choosing the best way to deal with a tile and so on - generally doing whatever it took to efficiently implement our method on a GPU. Finally we generalize the method to formulate a framework that can help guide the design of new architectures.
> - We will touch on this in our upcoming extended discussion on the context/literature review. However, since our method works for any convolutions, any SSM could benefit from it - it is just that if they are low-dimensional enough, the recurrent view could be faster, but our method can be applied as well. Asymptotically, assuming SISO (single-input-single-output), one would benefit from our method if the state dimension exceeds $\Omega(\log(L)^2)$. Besides SSMs, we hope the method has the potential to inspire future architectures (potentially transformers’ adaptations) and this is precisely why we propose the general framework mentioned in Section 3.4 and Appendix “The Flash Inference Framework”.
> - In our new revision we added the complete sweep mentioned in the Experiments section explicitly in the paper, together with some of the associated plots.
>
> Regarding the questions:
> - We did so above.
> - Following the integration of CUDA Graphs, we dropped the non-mixer times by a significant constant (since that part was dominated by CPU overhead). Following this revision, we now get very significant end-to-end improvements as well. Furthermore, even in the initial case we dropped the mixer part from the same order of magnitude with the MLPs to negligible - this then points to MLPs being a part to tackle next (say via structure matrix approaches). Regarding the improvements potentially becoming marginal, on the actual hyena architecture, we get at least $2.8\times$ improvements in the new experiments on every single hyperparameter combination (and up to almost $8\times$).
> - The reductions of Hybrid vs FFT are shown (numerically) for all experiments in the tables of Appendix Further Experimental Details, they are sometimes marginal, but sometimes get up to 10% extra performance. Regarding the non-mixer time being smaller for the eager/lazy counterparts in Figure 3c, this has been fixed via the usage of CUDA Graphs, but the initial cause was already covered in the caption as well as in Section 4.3, namely better mixers exposing CPU overhead. As mentioned above and in the general rebuttal, we got rid of this issue. We have added a discussion on this in Appendix D.2 (CPU overhead: the silent performance killer).

---

> > ### Comment · Reviewer_byxw · 2024-12-03
> > **Response to authors**
> >
> > Thanks for your clarifications. Given that I am not an expert on systems, I cannot evaluate the effectiveness of the practical parts regarding implementations, but the updated experiments seem extensive and solid. I decide to raise the score in light of authors' contributions to simulations.
> >
> > Another question: Can authors further explain why the (Flash) FFT methods are not evaluated on real-world Hyena experiments (i.e., Figure 2(a) only involves the proposed method and basic lazy/eager approaches)? Since it seems that FFT-based methods can also achieve comparable performance under the synthetic setting (Figure 3(c)), I am wondering the actual performance of FFTs when applied to real-world settings.

---

> > > ### Author Response · Authors · 2024-12-03
> > >
> > > Thank you!
> > >
> > > We didn't run Hyena non-hybrid simply because the hybrid was supposed to be by construction the best of all worlds and thought the qualitative differences between them are showed well enough in the synthetic experiments. There is nothing fundamentally stopping one from running the FFT-based methods on Hyena as well, though - and this should be easy to do once we release our code.
> > >
> > > For what it's worth, Hybrid always chooses one of the 2 FFT implementations for larger convolutions (as is expected given that they asymptotically do $O(U\log U)$ FLOPs per tile as opposed to $O(U^2)$), so there is already a big overlap between them in terms of what the approaches actually end up doing. That being sad, as far as your curiosity goes, one can expect the real-word (Hyena) mixer times to be very similar to the one in the synthetic - the code makes this clear since the tile-dealing part is effectively shared across the 2 kinds of experiments.

---

### Official Review · Reviewer_pkhg · 2024-11-04

**Soundness:** 4
**Presentation:** 3
**Contribution:** 3
**Rating:** 8
**Confidence:** 4

**Summary:**

This paper introduces "Flash Inference", a novel framework for efficient inference in long convolution sequence models (LCSMs) employing a tiling strategy inspired by relaxed polynomial interpolation . The authors propose an algorithm that achieves quasilinear $O(L \log^2_2 L)$ time complexity for exact inference, improving upon the quadratic complexity of standard approaches without resorting to any approximation techniques. The work specifically demonstrates the method's effectiveness on Hyena architectures, achieving significant end-to-end speedup and even more significant improvement within the position-mixing component of the model. Beyond LCSMs, the authors identify key properties that enable such speedups and propose a general framework to guide future architecture design. The proposed approach further reduces memory movement and enables parallel computation across layers.

**Strengths:**

1. Technical Innovation and Soundness:
- The paper presents a mathematically rigorous approach to improving inference efficiency, with clear proofs and careful analysis of complexity bounds
- The implementation details are thoroughly considered, including memory optimization and parallelization strategies
2. Practical Impact:
- The achieved speedups (1.6× end-to-end, 50× for position-mixing) are significant and well-documented
- The method is exact rather than approximate, maintaining model fidelity while improving performance
- The framework extends beyond just LCSMs, providing valuable insights for future architecture design
3. Experimental Validation:
- Comprehensive empirical evaluation across different hyper parameters parameters ($B$, $M$, $D$)
- Careful ablation studies of different $\tau$ implementations
- Clear breakdown of performance improvements and their sources

**Weaknesses:**

1. Technical Presentation:
- The notation could be improved for clarity, particularly in handling subscripts that simultaneously indicate sequence position, feature dimension, and layer. For example, you could consider using superscripts to indicate features (channels) and/or paranthesis around the subscripts/superscripts for the layers.
- the use of "$\mapsto$" notation on line 159 is imprecise; it should be $y\mapsto z$.
- Algorithm 1 should reference equation (3) for the definition of $\tau$ for better clarity.

2. Contextual Discussion:
- The discussion of approximate inference methods could be more nuanced. The statement about "defeating the purpose of using LCSMs instead of LTI SSMs" is overly strong, given that Hyena filters are intentionally underparameterized and some *implicit regularization* (convergence of the filters to ones that can be effectively represented by a truncated basis of complex exponential functions) is to be expected. Te reviewer suggests that the authors provide a more balanced discussion of the trade-offs between exact and approximate methods, considering the intentional underparameterization of Hyena filters and the potential benefits of implicit regularization.
- The introduction would benefit from a clearer distinction between prefilling and autoregressive generation phases in the complexity analysis. Elaborating on this could strengthen the clarity of how the method affects the performance of the system.
- The claim about "popular misconception" (line 109) regarding the realization theory of LTI systems could be better supported with appropriate citations. It is the reviewer belief that the this is not a misconception in the control theory literature as the problem of realization of LTI system has bee studied for more than 60 years  (e.g., Kalman and Ho 1960). Perhaps the authors could clarify on that.

3. Technical Specifics:
- The asymptotic complexity expressions should consistently specify the base of the logarithm ($log_2$)
- The parallelization discussion could benefit from more detailed analysis of the memory bandwidth implications
- The interesting extension to data-dependent filters, while covered in the appendix, could deserve more attention in the main text if supported by quality experiments (do we need input-dependent filters in LCSMs?)

**Questions:**

1. How does the performance of Flash Inference compare to existing methods when dealing with very long sequences (e.g., $L>$32K) in non synthetic scenarios? Are there specific challenges (other than memory) or additional optimizations possible at these scales?
2. Could you elaborate on how the tiling strategy might be adapted for architectures with more complex dependencies between layers? This seems particularly relevant for hybrid architectures that combine convolution and attention mechanisms.
2. The main limitation of the proposed method is certainly the linear growth of cache size with sequence lenght. How does this affect the practical benefits of exact vs approximate inference methods? In the approximate method, the cost of distillation can be amortized offline for a subsequent constant memory autoregressive generation. Moreover, one can trade additional memory (number of states in the SSM) for additional precision in representing the filters. Using a pre-trained LCSM, could you verify the accuracy-memory trade-off of the approximate vs exact method for practical sequence lengths?
3. Have you considered how this framework might be extended to handle structured sparsity patterns in the filters? This could potentially lead to additional efficiency gains while maintaining the theoretical guarantees.

---

> ### Author Response · Authors · 2024-11-23
> **First revision rebuttal**
>
> Thank you for the insightful comments! In our first revision we tried to cover all of them except for weakness 2 and question 3 that we are currently working on.
>
> Regarding the weaknesses:
>
> - W1:
>   - At line 163, we changed to hadamard notation to not make the channels explicit. Since the channels appear very sparingly, we decided to use the superscript for layers instead in hope that this makes the formulas more readable (and let channels/features be subscripts).
>   - We changed it to denote it being a definition.
>   - We updated the reference.
> - W2: We appreciate your points and will update our paper in the next revision to integrate a deeper and more nuanced discussion as requested. We wanted to make this first update to respond to the smaller points and update on our improved experiments but will be back to addressing this. Regarding the “popular misconception”, we were referencing the ML community where the bias mostly stems from these models having been proposed following low-dimensional LTI SSMs - however, we’ll change to something less subjective, we simply wanted to emphasize the fact that LCSMs' equivalent LTI SSM representation need not be low-dimensional (and thus, that general LCSM' inference is not settled through the recurrent approach).
> - W3:
>    - We considered it is common practice to not denote the base of the logarithm since the asymptotics of different bases stay the same: that is, $O(log_2(L))=O(log_{100}(L))$. We do not try to make any clear claims about the constant involved since that is a function of FFT implementation and more.
>    - We added to Appendix “Discussion on Memory” a deeper discussion on the relationship between across-layer parallelization and memory bandwidth.
>    - We did not intend to propose a new architecture ourselves, but with regards to whether we need data-dependency, we referenced two works (Zoology and Orchid) in the conclusion that show the potential of such methods - however they did not succeed yet in finding a causal alternative. We added a mention to these in the main body at line 235 as well.
>
> Regarding the questions:
> - In our revised version, we added experiments on higher context lengths (up to 128K and even 256K). There is nothing special about 32K, we only chose that plot since the CPU overhead was more stable across methods, but following the integration of CUDA Graphs, all the experiments have this property. As is expected, the higher the context length, the higher the mixer speedup. Any specific issues for certain context lengths would be due the FFT implementation (which past a certain point will struggle to fit the relevant sequences in the SRAM and thus incur a slowdown), but one would still expect this approach to be better than lazy/eager.
> - We have added this to Appendix “Framework Adaptations”, but in short the method can be applied to the relevant subset of layers that fit the framework. For example, one could use the same approach for Striped Hyena - even the across layer parallelization persists, but it can only happen on the relevant subset.
> - We agree this is a useful experiment to have and while it is difficult to measure accuracy tradeoffs since this is architecture and data-dependent, it certainly would be good to know from what SSM dimension onwards our method works better - theory would predict $\log^2(L)$ assuming SISO, but the constant could vary a lot. We are trying our best to get this experiment up in due time, but will mention at least the theoretical asymptotic. Regarding the linear growth of the cache, this seems fundamentally needed if one wants to use absolutely any filter length, but of course it is true that we could settle for particular structures.
> - We are actively looking into this. Interestingly, this would be easy to integrate if one assumes the filter to be sparse in the first few positions (and one could use some different layer for local behavior) - simply by dropping dealing with the small tiles. This would have the same theoretical complexity but probably improve the constant. Thank you for the suggestion!

---

> ### Author Response · Authors · 2024-11-24
> **Second Revision Rebuttal**
>
> We just updated to our second revision where we added an extended contextual discussion.
>
> Specifically, regarding the second weakness:
> - We dropped the strong statement and replaced it by just mentioning that we fundamentally change the class of models. We discuss in Appendix A the advantages of having a fast inference framework that does not rely on particular biases and give a more nuanced discussion on when it is desirable to choose our method compared to others that exploit certain structural properties (including SSMs).
> - We cover prefilling vs autoregressive stage in Appendix A.3. The only thing different from transformers is the eager nature of the prefill which lets us drop the first P activations (where P is the length of the prompt). We also mentioned how our improvements are most relevant when the prompt size does not dominate the number of generated tokens.
> - We dropped the claim altogether and rewrote that paragraph, redirecting to Appendix A for more background. We only state that there is a bijection between SSMs and LCSMs and that the equivalent SSM to our filter of interest could have too big of a dimension for the recurrent view to be efficient, while Flash Inference has a filter-independent cost.
>
> We hope this addressed your concerns!

---

### Official Review · Reviewer_qm9L · 2024-11-08

**Soundness:** 2
**Presentation:** 2
**Contribution:** 2
**Rating:** 6
**Confidence:** 3

**Summary:**

This paper proposes a method for speeding up Long Convolution Sequence Models (LCSM)'s exact inference to quasilinear time, identifies the key properties that make this possible, and proposes a general framework that exploits these. The proposed approach is inspired by relaxed polynomial interpolation and uses a tiling method that minimizes memory movement and enhances computation sharing, allowing near-complete parallelization across layers in the architecture’s position-mixing component. Through a proof-of-concept implementation on Hyena, we demonstrate up to a 1.6× improvement in end-to-end inference time and a 50× speedup in the position-mixing part.

**Strengths:**

1. The paper addresses a fundamental issue in sequence models, particularly long convolution sequence models (LCSMs) like Hyena, where inference time scales quadratically with sequence length. The proposed framework reduces this to quasilinear time, achieving significant improvements by leveraging a tiling-based approach inspired by relaxed polynomial interpolation.

2. Besides speed, the paper focuses on reducing memory movement, a bottleneck in handling large models. It suggests methods for activation storage optimization and discusses adjustments for memory-constrained hardware, making the work relevant for deployment.

3. The paper presents thorough empirical results that demonstrate the proposed method's effectiveness. It reports up to 1.6x end-to-end improvement in speed for Hyena and a 50x speedup within the convolutional mixer component, which provides concrete evidence of practical impact.

**Weaknesses:**

1. While the study is well-documented, the experiments focus primarily on synthetic setups and Hyena. Additional tests on a broader range of sequence models or with real-world tasks could further validate the framework’s generalizability.

2. The framework's efficiency relies on data-independent filters for optimal performance. Although data-dependent filters can be supported, doing so may lead to higher complexity or additional constraints. This could limit the application in cases where data-dependent filters are essential.

3. The framework assumes autoregressive causality, which could restrict its application to non-causal architectures. This point is mentioned briefly, but it would benefit from further exploration of how the framework could adapt to different model constraints.

**Questions:**

1. The paper suggests applicability beyond LCSMs. Could the authors provide additional empirical results or theoretical discussion on adapting the method to different architectures, such as transformers, beyond high-level discussion?

2. While data-independent filters optimize the framework's efficiency, data-dependent filters could require modifications. Could the authors elaborate on practical adaptations to support data-dependent filters effectively, and whether the framework maintains similar efficiency in such cases?

3. The work compares primarily with the eager and lazy approaches. How does the proposed framework perform against other efficient models or architectures, such as those using efficient transformer variants?

4. The paper assumes causality in model design. Are there potential modifications to the framework that could apply it to non-causal architectures? This could expand the impact and applicability of the proposed method.

---

> ### Author Response · Authors · 2024-11-23
> **First revision rebuttal**
>
> Thank you for acknowledging our contribution and for the comments! Our responses are below and we have revised our paper following them.
>
> Regarding the weaknesses:
> - We theoretically proved that the method applies to any LCSM (and LTI SSM for that matter). This includes by definition CKConv, FlexConv, H3, etc. Our synthetic ones already cover the general form of LCSMs: whether you place an MLP or some other kind of block, the mixer times stay the same and that is where our improvement comes from. We performed the Hyena experiments to prove this synthetic-to-complete-architecture transition, especially since Hyena is more involved than its counterparts. Regarding real-word tasks, could you please explain what sort of performance you refer to? Our method performs exact inference and the time does not depend on the weights, so the quality of a model stays the same. We also do not see how experimenting with any other LCSM architecture would add much - we only optimize the convolution part which we assume to be the same regardless of architecture (we even share the tile-handling code between our synthetic and hyena experiments), so we kindly ask you to let us know what concerns you have about the current setup.
> - We added an extra note at line 214 mentioning that data-dependent filters incur a factor of 2 (this was done implicitly at line 233 anyway). We covered at line 726 (Appendix C) how data-dependent filters simply require a different tiling that at each step will perform 2 FFT calls of the same size instead of one. Our whole theoretical analysis extends and given the very simple nature of the modification, one can easily expect the implementation to take at most twice the amount of time (it could be even faster than that for tiles that didn’t use to be memory or compute bound).
> - We added a discussion in the Appendix (Framework Adaptations) on this. In short, we make the absolute minimal assumptions needed for the model to well define autoregressive generation. In particular, one can accommodate non-causal encoding of a prompt for example or even chatbot-like interactions.
>
> Regarding the questions:
> - We have mentioned in Section 3.4 both why transformers cannot benefit from Flash Inference as well as a high-level description of what class of architectures our method can be applied to. The class is thoroughly defined in Appendix B. We could not identify any *existing* architectures beyond LCSMs that fit the description, but this is precisely the purpose of the generalization: to encourage people to try out such architectures - as suggested as a future direction in the second point of conclusions.
> - As mentioned above, we will incur at most a factor of $2$ efficiency cost practically (on the mixer side).
> - We could not find any well-defined way to compare inference for different architectures: is it for the same number of parameters or for the same compute training budget? Or for the same embedding dimension? LCSM’s parameters can be distributed very differently from those of transformers (since the attention layer has the Q, K, V matrices parameterize it whereas, for the same embedding dimension, the parameterization of the SISO convolutional filters could vary wildly in LCSMs), so as far as a fair direct comparison goes, we can only look at the theoretical asymptotics in $L$ (where attention takes $O(L^2)$ and we have $O(L\log^2(L))$).
> - We covered this above.

---

> > ### Comment · Reviewer_qm9L · 2024-11-24
> >
> > Thank the authors for the detailed response and clarification. I have no further questions and will raise my score.

---

### Official Review · Reviewer_xHkL · 2024-11-08

**Soundness:** 4
**Presentation:** 3
**Contribution:** 3
**Rating:** 8
**Confidence:** 4

**Summary:**

This work introduces a method to speed-up the autoregressive inference of long convolution sequence models (LCSMs) to near linear time. The approach is based on findings from relaxed polynomial interpolation’s literature, which is gratefully adapted to convolutions.
The resulting algorithm results in important speeds-up both for the position-mixing components and end-do-end inference.

**Strengths:**

- The method is novel and offers interesting improvements in the inference speed of LCSMs.

- The paper offers interesting perspectives that could be used for the design of more efficient (causal, input-dependent) LCSMs in the future.

**Weaknesses:**

- The main weakness of the paper is that the presentation, design decisions and final implementation of the method remains quite abstract, even after reading the paper multiple times. Given that the paper presents an inference strategy, it should be feasible to have an stand-alone implementation (at least for one layer) incorporated in the Appendix of the paper. This would give clarity to the final, concrete version of the algorithm.

- Next, I feel that the presentation of the paper could be improved. For example, the function $\tau$ –which is crucial for the method– remains undefined through the whole body of the paper up to Sec. 4.2, where it is briefly and loosely defined. It is mentioned that 7 possible versions were tested, but only 4 were mention-worthy. Then, it is mentioned that some are used, but it is not specified in which ranges and under which parameters one is preferred over the other. Given that this is core to the method, this should definitely be improved.

**Questions:**

### Additional questions and observations

- Contribution 1. This is true, however, for clarity, I would recommend that the authors mention Laughing Hyena before making this claim to put in context that that method is not exact.

- Line 233. There’s a typo here that changes the whole meaning of the sentence. Please fix.

 ### Conclusion

While I believe that the core contributions of this paper are very valuable, I am not really convinced by the current presentation of the paper. I would recommend the authors to improve readability and make specific the different design decisions that make the paper.

Do note that the contribution of this paper is rather an algorithmic one. Yet, the current very abstract depiction of the method makes it unnecessarily difficult to implement and reproduce. I am, therefore, only able to provide this paper a “weak acceptance” at this time. However, that if the authors were to improve over the weaknesses outlined here, I would be more than happy to increase my score.

---

> ### Author Response · Authors · 2024-11-14
>
> Thank you for the thoughtful comments!
>
> We are happy to try and integrate them, but before doing so we have a few questions to make sure we address the right issues:
> - Regarding the first weakness, we do provide Algorithm 2 which is the complete algorithm for an end-to-end LCSM model as well as Algorithm 1 which deals solely with one convolution layer. We could, if you find this to address your problem, add a python implementation for one layer (it would be mostly the same as Algorithm 1 but with an extra MLP and the $\tau$ function implementation based on FFT). Otherwise, could you please pinpoint which parts of Algorithm 2 are too abstract? Could you also please detail what design decisions you have in mind? We are not aware of any implicit decisions, but are happy to add more details about any part.
> - For the second weakness
>    - The function $\tau$ has first been defined at lines 261-262 (Eq. 3) for the case of relaxed polynomial interpolation and then overwritten for the general case at line 338. We intentionally use the same symbol since it has the same function, just across more dimensions, so that it is easier to follow the transition from Algorithm 1 to Algorithm 2. It is true that the first definition is as part of Lemma 1's statement: we are happy to define it before we state the Lemma to avoid any confusions.
>    - We will add in the Appendix the other 3 implementations of $\tau$, but regarding how we choose between them: this is covered at lines 478-479. Specifically, we profile each implementation for each batch dimension, embedding dimensions, number of layers and tile size and automatically choose the best one for each iteration. There was indeed a typo at the end of the sentence: we should've referenced Figure 3a which shows the runtimes for the four implementations for varying tile sizes. We can provide the specific implementation choice for each (B, D, M, U) quadruple in the Appendix if that helps, but generally the pattern is as showed in Figure 3a: the quadratic implementations are better for small tiles and then the FFT ones dominate with the torch vs Flash implementations sometimes flipping.
>
> Regarding the first question, thank you for the suggestion, we will add it. As for the second question, what we meant is that van der Hoeven's approach could be used as well, but since $\rho$ is known ahead of time we can do better. We hope the following restatement is clearer:
> > Whereas one could simply use the more general approach of (van der Hoeven, 1997), we can further take advantage of ρ being known ahead of time to get a twice-faster and slightly simpler algorithm
>
> The reason for the slightly more abstract view is simply to cover all LCSMs at once. We are happy to address any specific design choice you may have in mind and implement any actionable advice.

---

> > ### Comment · Reviewer_xHkL · 2024-11-24
> >
> > Dear authors,
> >
> > Apologies for my late reply. I just went through the paper again. I feel that in the current version my limitations have been properly addressed. It is not necessary to have the implementation on the paper. As a final question, are you planing on releasing your code at some point? Doing so would be very useful and probably improve the reach and impact of your work.
> >
> > With no concerns left, I am raising my score.
> >
> > Best,
> >
> > The reviewer

---

> > > ### Author Response · Authors · 2024-11-24
> > >
> > > No need to apologize, thank you for the review and for confirming we addressed your concerns!
> > >
> > > Yes, we are surely going to release our code, we are working towards cleaning it up and writing the readme (and anonymizing it), but it will be available.

---

### Official Review · Reviewer_iRJb · 2024-11-08

**Soundness:** 4
**Presentation:** 2
**Contribution:** 2
**Rating:** 5
**Confidence:** 4

**Summary:**

This paper proposes more efficient algorithms for auto-aggressive inference of long convolutional sequence models (LCSMs). The aggregate running time on a sequence of length $L$ is reduced from $O(L^2)$ to $O(L \log^2 L)$, and the actual wall clock time of the implemented algorithm reflects substantial increases in efficiency.

**Strengths:**

The paper's main strength is that it is technically sound and novel, and addresses the problem it aims to solve.
- The technical writing is clear and is helped by the inclusion of helpful graphics and rigorous algorithm boxes.
- Many considerations and variants of the core algorithm are proposed.
- An actual implementation is provided and all variants and baselines are benchmarked empirically.

There is a conception that long convolutions cannot be implemented efficiently in autoregressive inference settings, and so I do think that this paper presents an original algorithmic contribution.

**Weaknesses:**

While the paper provides a technical contribution, the paper's main weakness is that of significance and direction with respect to the broader field; it aims to solve a problem that I believe does not need solving. Correspondingly, the papers writing (in terms of positioning and related works / baselines) could also use improvement.

- The paper's related work is sparse and I think it is important to present the lineage of these models more carefully. The original (depth separable) LCSMs were independently developed by two lines of work: the implicit convolution (CKConv and FlexConv) line of work, and the SSM line of work (LSSL, S4, DSS/S4D, H3, MEGA/Megalodon, and many more). Even though I understand why the paper deliberately puts emphasis on LCSMs that are not SSMs, because this is where its results are most applicable to, the positioning is at times misleading (e.g. in paragraph 2 of the introduction, where LCSMs are implicitly defined as non-SSM models, even though LTI SSMs are in fact the original models that popularized LCSMs).
- It is odd that the paper is heavily anchored around the Hyena architecture, even though it does not actually do anything Hyena-specific. The experiments don't use an actual trained model or look at empirical performance, only the speed of an architecture, for which any model with a LCSM convolution (e.g. H3, to which Hyena is equivalent except for the choice of convolution kernel) could equivalently be substituted into the writing without changing the algorithmic results.
- A distinction is made that low-dimensional LTI SSMs cannot represent general LCSMs, which is true and where the paper's potential benefits can come from. However, current understanding of LCSMs is that they need to be defined using certain priors (e.g. baking in exponential decay as in SGConv and Hyena) that essentially are similar to the priors imposed by low-dimensional SSMs. Empirically, it is known that there is little difference between these models (e.g. https://arxiv.org/abs/2312.00678v1 Fig 1, which claims that H3 in fact performs better than Hyena, where the only difference is the choice of convolution kernel parameterization). Thus a major weakness of this paper is that it is currently only applicable to models that have better alternatives empirically.

**Questions:**

I think the paper needs to position with respect to other types of LCSMs with more nuance. For example, it could benchmark how the proposed algorithm for general LCSMs compares to the inference speed of recurrent LCSMs; e.g. for a given sequence length $L$, at what recurrent state size does the speed of a recurrent SSM cross over the speed of the proposed algorithm? This would at least provide some more useful context for the reader interested in the pure algorithmic aspects.

However, overall, in order for this paper to be valuable to the machine learning community, it should be applied to actual models, and current understanding of LCSMs is that general LCSMs essentially do not benefit over low-dimensional LTI SSMs. The contributions are interesting from a purely algorithmic perspective, and the paper does suggest potential extensions beyond time-invariant convolutions. However, it is not clear whether this has any chance of being extended into performant models - no actual downstream model is proposed. Thus, I think that a conference such as ICLR is not the most appropriate venue for such a submission without anchoring the algorithmic contributions to real models. In order to increase my score substantially, I think the paper needs to show a practical utility, for example either
- Showing that there exist classes of models that the algorithm can be applied to that are not dominated by others (e.g. showing that some non-SSM LCSM model is actually faster or more performant than an equivalent SSM)
- Showing that there exist data-dependent extensions that outperform data-independent convolutions in modeling performance

---

> ### Author Response · Authors · 2024-11-14
>
> We thank the reviewer for the thorough comments and for acknowledging our technical contribution!
>
> Regarding the weaknesses:
> - While we are happy to add more context (and thank you for the pointers, have missed FlexConv), the only SSM that makes use of the convolutional view (because it has a meaningfully large state dimension) is S4 - all the others exploit a small state dimension which LCSMs do not assume. We will add a section in the appendix on the connection between LCSMs and SSMs, but as stated in paragraph 2, we acknowledge that any convolution has an equivalent LTI SSM - it’s the dimension of that state we emphasize as being the core difference. The reason we did not delve deeper into this is simply because the purpose of the paper is not to argue for the use of LCSMs, but merely to provide an efficient way of serving them - an open problem that was also approached in Laughing Hyena Distillery.
>
> - Regarding the anchoring around Hyena, we will drop (or broaden) the references at lines 17, 43, 213 and 530. Thank you for the pointer - we thought Hyena is the most well known of the LCSMs and wanted to make explicit which parts apply to it as well. Moreover, the closest work to ours in terms of problem setup is Laughing Hyena Distillery.
> Do you mind elaborating what performance you would find meaningful to evaluate? Since we do exact inference, we only claim a certain speed-up: anything else related to the statistical quality of the model stays unchanged.
> Regarding why we tried only Hyena: is there any architecture you believe running extra experiments on would add any value? The reasoning was that the synthetic setting should cover all LCSMs and Hyena is just an extra example. Your last point about the method being generic is true and our synthetic setting, together with Algorithm 2 are not specific to Hyena in any way.
>
> - Firstly, Fig.1 of the linked paper does not seem to have any data on H3/Hyena. Secondly, and more importantly, it is very challenging to claim complete superiority of an architecture over another, especially in the opposite direction of representational power, simply because one cannot consider all the ways that module could be used: performance could vary a lot depending on use case and integration. Our method speeds up a long convolution layer, which is generic enough to be considered a useful building block for yet-unexplored scenarios. It could be that H3>Hyena by themselves and hybrids of attention with H3 are worse than hybrids of attention with Hyena. To give a precise example, EVO (which is based on Striped Hyena) is the current state-of-the-art in DNA modeling. Moreover, one could always change the biases of the convolutional filters: the main issue is that of training stability, which is itself an open problem. Our method serves as a tool readily usable for any such method. We believe it is useful for the community to be aware of a fast inference alternative for them to not constrain their research to low-dimensional SSMs because of computational considerations.
>
> Regarding the questions:
> - EVO (based on striped hyena) is an example of a long convolution-based model that is more performant than the equivalent SSMs (EVO tried mamba which failed due to numerical instability). We unfortunately cannot easily test how much of the accuracy is lost when distilling in due time, but the model is trained in the convolutional, not SSM view. S4, although primarily viewed as an SSM, is still SOTA on the Image section of the Long Range Arena tasks and requires a dimension equal to the sequence length.
> - Orchid and Zoology specifically show the representational advantages of data-dependent convolutions to synthetic tasks of interest - both of which suggest non-causal architectures which outperform SSM-based counterparts. It is an open problem to find a good *causal* architectural choice that is stable to training, but it is clear that the added representation power closes a gap.
>
> Regarding the broader positioning, we believe it is important to not overlook one main contribution: that of defining exact properties that allow for similar improvements to be applied to other architectures. We consider this is a good reason for the paper to be published at a conference such as ICLR since people can take these considerations into account when designing the next generations of architectures. While this is touched just briefly in the main paper due to limited space, we have a complete analysis in the appendix. Furthermore, we hope that the technical part of our method inspires people to think differently about similar problems that may arise in their practical research. Finally, we would also appreciate if you detailed what you meant through “real models”, since we do have a realistic Hyena implementation which is a real model.

---

> ### Comment · Reviewer_iRJb · 2024-11-14
> **clarification**
>
> I think there is a fundamental confusion here about definitions. Do we agree or disagree: by your definition of LCSM (section 2.3), all SSMs (henceforth assumed to be LTI and SISO unless otherwise qualified) are LCSMs?
>
> > the only SSM that makes use of the convolutional view (because it has a meaningfully large state dimension) is S4 - all the others exploit a small state dimension which LCSMs do not assume.
>
> First, all SSMs mentioned in my original review ("LSSL, S4, DSS/S4D, H3, MEGA/Megalodon, and many more") do use the convolutional view. This has nothing to do with state size.
>
> > but as stated in paragraph 2, we acknowledge that any convolution has an equivalent LTI SSM - it’s the dimension of that state we emphasize as being the core difference
>
> Second, my claim is not about any convolution having an equivalent LTI SSM, but the converse: all SSMs are by definition LCSMs.
>
> These claims have nothing to do with the state size of the SSMs, for now. Let's get on the same page about this before I respond to the rest.

---

> > ### Author Response · Authors · 2024-11-15
> >
> > Regarding the converse, yes we are on the same page that all LTI SSMs are also LCSMs, and the paper does say this in its current form in paragraph 2:
> > > while any linear-time invariant (LTI) SSM has an equivalent convolutional filter
> >
> > And yes, you are right, that is a mistake on our end in the rebuttal (primarily driven by most of those models using the recurrent view for inference). The convolutional view does have to do with the state dimension in that choosing the parallel scan or FFT for training is a function of how large the state is. But we do agree that all SSMs are also LCSMs. And implicitly, any (LTI) SSM could use the convolutional view if that is preferable. We will more thoroughly go through the enumerated papers in our discussion, but feel free to proceed with responding to the rest of the rebuttal since we are on the same page.

---

> > > ### Comment · Reviewer_iRJb · 2024-11-15
> > > **Response 1/2**
> > >
> > > Okay great. With that in agreement, the core issue is that the paper has many confusing or even wrong statements about LCSMs. For example, the relationship between SSMs and LCSMs is not clear and often used inconsistently (e.g. it is often implied that "LCSM" does not include SSMs, in contrast to what we just agreed upon). This makes the positioning of the submission (e.g. the context with respect to prior work and related models) quite confusing and often incorrect, and also raises many unanswered questions that are important to justify the significance and utility of this work (e.g. what are the tradeoffs of the proposed algorithm with other alternatives).
> > >
> > > 1. As mentioned in the original review, paragraph 2 of the intro (and much of the rest of the paper) does not accurately portray the lineage of LCSMs: e.g. SSMs are the models that popularized LCSMs, while here (and in many other places) the term "LCSM" excludes them and seems implicitly to be referring to "LCSMs that cannot be represented by a low dimensional SSM". It is fine to define a new term that encapsulates this but it should be made more precise.
> > >
> > > 2. There are many incorrect statements perpetuated about LCSMs throughout the paper, and repeated in this discussion so far. One example is about the aforementioned related work, which are in fact all LCSMs that are actually computed using FFT convolutions during training. Similarly, the authors' last response said "The convolutional view does have to do with the state dimension in that choosing the parallel scan or FFT for training is a function of how large the state is." This is incorrect: whether or not SSMs choose to use convolutions during training is a function of (1) whether they are LTI or gated/selective (2) whether they are SISO or MIMO. All LTI+SISO SSMs (including all aforementioned models, and more such as Liquid-S4, some versions of RWKV, etc) use convolutions during training for even very small state sizes, as FFT is generally much faster than scan on GPUs. In other words, there is a large class of models that are actually pure LCSMs by definition, that are not described accurately. There are many more technically incorrect statements, some of which are touched on in the below points, but this list is non-exhaustive.
> > >
> > > 3. It is important to acknowledge that while the proposed algorithm applies to all LCSMs, however, there exist many subclasses of LCSMs with alternative efficient inference algorithms. Correspondingly, I think that benchmarking the inference speed of the proposed *general LCSM* algorithm against other families of *efficient LCSMs* is critical to the submission. For example from the perspective of the broader community, one would want to know when should they use an SSM or when should they aim for a different class of models at the cost of slower inference (and how much slower). Or maybe the proposed algorithm is actually faster than a low dimensional SSM! - the reader doesn't know because it's not benchmarked and not discussed. Other families of efficient LCSMs exist, such as MultiresConv (https://arxiv.org/abs/2305.01638).
> > >
> > > 4. Several areas of the paper implicitly use "LCSM" to imply "a fully expressive convolution filter" and "SSM" to imply "a less expressive low dimensional subclass". But this is quite an arbitrary distinction, and also not technically sound. First, LCSMs are by definition (section 2.3) models that create long convolution filters through a smaller set of parameters. Thus *every* LCSM is not fully general in expressivity (a point which the paper often makes confusing statements about), and the real question is what inductive biases are imposed through a particular parameterization. Second, the authors acknowledge that in the limit of state size, SSMs become fully expressive LCSMs, so why make an arbitrary distinction between them - the proposed algorithm can equally be applied to sped up high-dimensional SSMs, which are simply one particular parameterization of an LCSM.
> > >
> > > 5. I think that the anchoring around one particular instantiation of LCSM detracts from the message of the paper.

---

> ### Comment · Reviewer_iRJb · 2024-11-15
> **Response 2/2**
>
> Some of my other points are about the utility of such models. As of right now, there are no results that suggest that there exist general LCSMs that outperform efficient LCSMs.
>
> 6. In all benchmarks where controlled comparisons exist (e.g. not including the EVO model, where no third party reproductions exist or ablations between the Hyena parameterization vs other LCSM parameterizations), the performance of non-SSM LCSMs and SSMs are essentially the same.
>
> 7. E.g. as the authors note "S4, although primarily viewed as an SSM, is still SOTA on the Image section of the Long Range Arena tasks and requires a dimension equal to the sequence length." Actually S4 and variants (MEGA, etc.) all use a low dimensional SSM, and the majority of methods near SOTA on LRA are based on low-dimensional SSMs or equivalent. I will also point out that the authors' phrasing "although primarily viewed as an SSM" once again implies a false dichotomy between SSMs and LCSMs that is being perpetuated. *SSMs are simply a class of LCSM with additional properties.*
>
> 8. There is in fact a technical reason for this, which was touched on in the original review. As mentioned in point 4 above, *all* LCSMs have restricted expressivity. This is essentially tight, in the sense that any LCSM with $P$ parameters requires $O(PL)$ time to construct its convolution filter, so there is a fundamental tradeoff in the expressivity <-> training time of LCSMs; both implicit convolution models like CKConv/Hyena as well as SSMs attain this bound. So there is no intrinsic expressivity advantage to using a non-SSM LCSM over an SSM (unlike what is often implied by the paper; for example, pointing out that general convolutions cannot be represented by a low-dimensional SSM is misleading, as they cannot be represented by any other finite-parameter class of LCSM either). Thus the question to ask is not about expressivity but about whether a particular parameterization carries a helpful inductive bias. However, the best non-efficient LCSMs such as Hyena and SGConv actually bake in priors like exponential decay which intentionally give them inductive biases *more similar* to low-dimensional SSMs. Overall, current understanding of the community is that performant LCSMs are empirically indistinguishable from low-dimensional SSMs.
>
> Thus I have significant doubts about the practical utility of the proposed algorithm for fundamental technical reasons. However, I understand that these empirical points are not comprehensive.
>
> Even if we ignore these concerns about the applicability, which I am willing to accept on the grounds that the algorithmic contribution is solid and can be the potential basis for future work, I cannot increase my score without a significant improvement in presentation. In particular, it should discuss the expressivity of LCSMs with more accuracy/nuance and present the taxonomy of various LCSMs and associated tradeoffs much more carefully. **These questions are central to the context, significance, and applicability of this work.** Concretely, for example, it should benchmark the proposed algorithm for general LCSMs against the inference of recurrent LCSMs (and ideally other LCSMs e.g. based on dilated convolutions). Note that I am not saying that the paper needs to show that its algorithm is faster, but that it should show the crossover point and discuss the theoretical and practical tradeoffs accurately, to better position the proposed contributions within the broader space of closely related literature.

---

> > ### Author Response · Authors · 2024-11-24
> > **Post second revision response**
> >
> > We have updated the paper to try to integrate your comments and in particular we thank you for the suggestion to not distinguish between LCSMs and low-dimensional SSMs as we think the paper's context is now much clearer and complete. We hope that our current revision properly addresses most of your concerns (we changed lines 103-112, 197-199 and added Appendix A).
> >
> > However, regarding your points that we did not cover in our previous response:
> > - P5: This is a point we had responded to and said we would address (and we did in the meantime)
> > - P6: Even if this was the case, models like EVO make for an immediate use-case of our method. One could postulate that an independent implementation attempting to use LTI SSMs would outperform or match it, but practically speaking, at this point, if one is to be interested in using the best available model, this is a hurdle to get past where our method can help.
> > - P7: This point overlaps partially with P1 and P2. We want to reemphasize this relates to the rebuttal and not the paper and that "primarily viewed as an SSM" is not supposed to entail a dichotomy. Primarily suggests that it could be viewed differently - and indeed it can. Any LCSM can be "viewed" as an SSM and vice-versa. The SSM view comes with parallel-scan and recurrent inference whereas the LCSM view comes with FFT and our method. One can choose either of them for either of the stages, but realistically to apply one, a temporary change of perspective is needed (while keeping the model unchanged). We did not mean to say more than this.
> > - P8: Besides the filter-computation argument, this seems to repeat P4 that we addressed. As for the argument, it seems like this makes an element-wise assumption of the sort $\rho_t=f(\theta, pos_t)$? We do not see any good reason to constrain oneself to this form (and thus no basis for the $O(PL)$ claim); In particular, [1] has $P=L$ and performs no extra computation. However, one could envision some other way of building the filter (say by having positions use different subsets of the parameters).
> >
> > We rewrote Appendix A in a way that we believe make milder claims than in the rebuttal (specifically since, as per your useful suggestion, we now treat SSMs and LCSMs equally). However, we wanted to clearly respond to the points to emphasize consistency with our notation and ask for a reevaluation to make sure we covered all your theoretical points. As mentioned in the general rebuttal, we are doing our best to perform experiments measuring the SSM dimension threshold where our method becomes desirable in due time.
> >
> > Finally, regarding your main point about the problem not needing solving - while we do not know yet the threshold dimension, theory predicts it to be $O(\log^2(L))$ which could actually make the method practical for some "average"-dimensional SSMs (such as S4's 512 dimension). Furthermore, even if we do not currently manage to get the threshold close enough to those of SSM dimensions, the asymptotical analysis suggests that further research in improving the implementation could be enough. We hope you recognize this as a stronger point regarding the relevancy of our work. This adds to other contributions such as allowing for data-dependent filters and phrasing a more general framework that could guide future architecture design.
> >
> > [1] Daniel Y Fu et al. Simple hardware-efficient long convolutions for sequence modeling, International Conference on Machine Learning. PMLR, 2023.

---

> > > ### Comment · Reviewer_iRJb · 2024-11-29
> > > **Response**
> > >
> > > I appreciate the authors' responses. I am unsure how to resolve our disagreements, as they appear to be due to fundamentally different understandings of what LCSMs are. I will try to reply to a few of these points.
> > >
> > > > Do we agree or not that low-dimensional SSMs are not fully expressive in terms of the convolutions they can represent?
> > >
> > > Yes, SSMs cannot represent all convolutions. *But neither can any other class of LCSMs*, so I think it is misleading to make statements such as this.
> > >
> > > I think that there is an implicit swapping of qualifiers here. By definition, an LCSM is an implicit parameterization of a convolution filter $\rho = f(\theta)$ (Sec 2.3). Let $T$ be the set of all possible LCSM parameterizations $f(\theta)$. Let $S$ be any strict subset of $T$ (e.g. the class of efficient inference LCSMs, or the class of SSMs, etc). Then indeed $S$ is clearly "less expressive" than $T$.
> > >
> > > However, in reality a model always has to choose a *specific* parameterization $f(\theta)$ first, rather than qualifying over entire sets of different parameterizations. Upon choosing a parameterization, there are little differences between most reasonable models. That is, a particular LCSM $f(\theta) \in T \setminus S$ is no more expressive than a different $g(\theta') \in S$ (e.g. supposing that $\theta$ and $\theta'$ have roughly the same number of parameters).
> > >
> > > An example of language where this is used in a subtly inaccurate way is (Sec 2.3.2)
> > > > More importantly, this approximation represents a projection to a potentially smaller space of models
> > >
> > > I think that typically the paper compares "SSMs" and "LCSMs" in the $S \subset T$ sense above, leading to notions like "smaller space of models" here. But the correct perspective is to compare $f(\theta)$ vs. $g(\theta')$. That is, fixing a parameterization of a LCSM $f$ (e.g. Hyena), the paper by Massaroti et al. projects it to a *different* (not smaller) LCSM parameterization $g$ (e.g. low-dimensional SSM). $g$ may be equally expressive as $f$ but performance may go down because the intersection of the ranges of $f$ and $g$ is smaller than the range of $g$. However the optimal solution for $g$ likely lies outside this intersection and so training from scratch would lead to equally performant models (matching for parameter count and hence computation speed).
> > >
> > > I do agree with the author's sentiment that, conditioned on trained models already existing for inefficient LCSM parameterizations, the proposed inference algorithm presents a viable approach for using these models without projecting to a different class of LCSMs.
> > >
> > >
> > > > Any LCSM can be "viewed" as an SSM and vice-versa. The SSM view comes with parallel-scan and recurrent inference whereas the LCSM view comes with FFT and our method.
> > >
> > > I find this perspective very confusing. I think that statements like these imply a dichotomy between SSM and LCSM, e.g. one is saying that "given an SSM model, one has to take a different perspective and view it as an LCSM in order to use FFT convolution". It also implies that "SSM" is fundamentally associated with recurrence/scan and not convolution. I don't think this perspective is accurate. As we have agreed on, SSM is a model parameterization that is a specific class of LCSM, and thus "SSM" itself is fundamentally associated with convolution. No change of perspective from "SSM" to "LCSM" is needed to use convolutions.
> > >
> > > To be more explicit, since I'm no longer sure if we agree about definitions: by this paper's definition of LCSM in Sec 2.3, an SSM is an LCSM $\rho = f(\theta)$ where $\theta = (A, B, C)$ and $f(\theta) = (CA^i B)_{i \ge 0}$ (eq (7) of https://arxiv.org/abs/2110.13985 or eq (5) of https://arxiv.org/abs/2111.00396). Note that this is a technical instantiation of LCSM by the paper's definition, and has no reference to recurrence at all.
> > >
> > > Another prominent example of this perspective, which has been brought up multiple times in this discussion already, is the way LCSMs are being implicitly defined throughout the paper through examples and citation patterns. Paragraph 2 (and Sec 2.3 to a lesser extent) provides representative examples of prior LCSM models, yet contains no SSMs. I cannot conceive of a good reason why these citations (which implicitly define the term "LCSM" by example, and thus is important to be comprehensive) exclude *the original class of LCSMs that popularized LCSMs*. This is another example of the "dichotomy" where SSM and LCSM implicitly are treated as separate classes of models.
> > >
> > > Overall while I think that the proposed algorithm is a solid contribution, I am worried that the context and lineage of LCSMs has not been described accurately, in a way that can potentially be harmful to the community. Similarly I think that benchmarks against efficient LCSMs are a necessity and not a nice-to-have, as these baselines are fundamental to the original motivations and very origins of LCSMs themselves. Other reviewers have expressed similar opinions about the baselines.

---

> > > > ### Author Response · Authors · 2024-12-03
> > > >
> > > > Thank you for the response! Regarding your points:
> > > >
> > > > - There is no qualifier swapping: as emphasized in the very paragraph you are responding to, our definition does not require $\rho$ to take that form, as the filters are just *potentially* underparameterized (line 44) as well as *often* underparameterized (line 161, where we say $\rho=f(\theta)$ in the underparameterized case). We believe there may be a misunderstanding regarding these references. Our rebuttal explicitly clarified that we do not assume underparameterization as it is not central to our work.
> > > >    - To further support this, we provided a reference that uses explicitly parameterized filters (albeit with additional regularization).
> > > >    - We also agree to your point about a *particular* parameterization not necessarily being more expressive (if anything one would expect it to also be unable to represent some of low-dimensional SSMs parameterized in the same count of parameters). However, we did make the exact same point in our rebuttal where we mentioned "depending on the choice of parameterization" and "Any one filter has at least some (*likely unrealistic*) parameterizations that would cover it".
> > > >     - At least as far as the rebuttal goes, we do not see any sign of not being on the same page; it is just that our work intentionally approaches general filters rather than particular parameterizations. It is within the scope of the contextual discussion of Appendix A to help decide whether this extra freedom earns one anything, and at least as far as the reference we gave goes, this is still an open question.
> > > >        - In particular, we believe lines 637-643 address this in detail. We recognize that time constraints might have limited a full review of the revision, but we believe these lines precisely clarify the points raised.
> > > > - We are happy to revise the wording in the final version to "potentially different," as it is true that the space of low-dimensional LTI SSMs might include filters not covered by the Hyena parameterization. Our intention was to highlight the risk of such a projection, which remains a valid concern.
> > > > - Regarding the comment on the perspectives, We see this as largely a matter of intuition, which can be subjective. To be precise:
> > > >    - We respectfully disagree with <No change of perspective from "SSM" to "LCSM" is needed to use convolutions>, since the change of perspective happens when one defines $\rho_i=CA^iB$, abstracting $A, B$ and $C$ into the convolutional representation. While these parameters could be viewed as simply parameterizing $\rho$, they hold deeper meaning in the recurrent view. In our opinion, this distinction reflects the broader utility of $A$, $B$, and $C$ beyond just parameterization.
> > > >      - This discussion seems to hinge on differing interpretations of "perspective". Nevertheless, we have been careful to state that SSMs and LCSMs are equivalent, while associating each name with its corresponding perspective for clarity.
> > > > - We will do our best to put all citations together in the main body *as well*, but we already did so in the Appendix. This has more to do with space and citation format. We extensively touched on the relationship of works (including implicit vs explicit) in the Appendix and the only reason that is not already in the main body is the space limit. We will drop swap some content to get more of it in the main text, but even right now we do refer to the Appendix for the extended discussion.
> > > > - As mentioned, we are trying to run those experiments. However, we want to emphasize that there is a significant theoretical contribution to our work as well (if not the primary contribution). This is a meaningful result even in absence of empirical validation, as we stated the theoretical result that one should prefer our method for LTI SSMs of state dimension $\Omega(\log^2(L))$ - the constant is a function of implementation, architecture and hardware. Our current experiments, together with the systems improvements, are meant to show the potential of the method.
> > > >    - As for why doing these experiments takes considerable effort: we need to profile different tiling mechanisms in the end-to-end architecture for each convolution implementation in order to be able to use our hybrid method - this requires extra code (on top of changing the architecture and baselines).

---

> ### Author Response · Authors · 2024-11-23
> **Post-first revision partial response**
>
> Thank you for the comments! We are happy to try and integrate them in our second revision where we will focus on a deeper contextual discussion. However, before doing so we were hoping to make sure we are on the same page about the current revision.
>
> Currently, SSMs are barely mentioned in the it and these are the only places we could find that we thought can be misinterpreted or confusing and our (performed) fixes:
> - At line 108 we could make it explicit that we mean parallel-scan-based training would be quadratic - we expected the word *LTI SSM approach* to convey this.
> - At line 197-198, we will re-emphasize *low-dimensional*.
>
> While we agreed and acknowledge the first rebuttal error (regarding stating S4 being the only one using FFT for training), we are not aware of any other places that you refer to through <incorrect statement perpetuated throughout the paper>, <often implied that "LCSM" does not include SSMs">, <many more technically incorrect statements>. We agree with the point that we could and should add more nuance and are working on that, but in the current form, the paper simply does not try to cover the differences and we do not see what technically incorrect statements you are referring to. Do you mind pointing us to what other places you believe we should fix in the current version? It will be helpful for our upcoming revision.
>
> Regarding points:
> - P1: We never meant to refer to LCSMs as "LCSMs that cannot be represented by a low dimensional SSM", but rather to emphasize that LCSMs contain strictly more than low-dimensional SSMs. Could you please pinpoint which part of the paper gave this impression of a dichotomy?
> - P2: We were assuming to be in the context you mentioned (LTI and SISO). Granted this, it is a function of state dimension whether one chooses FFT or parallel-scan, both in theory and in practice. It is true that even the theory predicts the threshold to be at $\log(L)$ which once accounted for constant factors can be (and truly is) very small. However, we do not see how our statement is incorrect (again, assuming LTI and SISO as the reviewer suggested to assume) - we never defined what low means precisely because we only mean something below a certain threshold, which could indeed be very low. And while we agree to the point that for such SSM dimensions, one would prefer to use the convolutional view during training (and we did acknowledged this mistake in the rebuttal), as well as to the value of emphasizing it in our work, we are not aware of having made such claims in the paper (and thus do not see how this was perpetuated in the discussion).
> - P3: we agree this could be useful to know and we will be trying to benchmark them, but since we need to have a fair comparison we need to integrate them in our current implementation.  We will do our best to do so in due time.
> - P4: the only point where we did not mention *low-dimensional* was lines 197-198 mentioned above. Do we agree or not that *low-dimensional* SSMs are not fully expressive in terms of the convolutions they can represent? While it is common to use underparameterized forms of LCSMs our definition uses the word *often* (line 160) to not constraint to that setting. In the introduction as well, at line 44, we mention they are just *potentially* underparameterized. So in the current state we do believe LCSMs could also be explicitly parameterized if one desires to do so (and, our method does not make any assumptions on how they are computed). This has (up to regularization) even been attempted by [1].
>    - Furthermore, even if they were underparameterized (which again was not our intention to mean, nor do we think we did so, but we can emphasize this to avoid confusion), depending on the choice of parameterization, one could easily extend to filters that are not representable via low-dimensional SSMs. Any one filter has at least some (likely unrealistic) parameterizations that would cover it. That parameterization corresponds to a choice of bias.
>
> However, we agree that there is no need to make an arbitrary distinction and that the method also applies to low-dimensional LTI SSMs. The reason for the distinction was simply because the low-dimensional SSMs already have a good (time and memory efficient) inference alternative. We appreciate the point that one could use our method for say average-dimensional SSMs where to determine the right dimensional threshold we would need the experiment you mentioned; we will touch on this.
>
>
> [1] Daniel Y Fu et al. Simple hardware-efficient long convolutions for sequence modeling, International Conference on Machine Learning. PMLR, 2023.

---

### Official Review · Reviewer_zv8x · 2024-11-11

**Soundness:** 4
**Presentation:** 3
**Contribution:** 3
**Rating:** 8
**Confidence:** 4

**Summary:**

This paper introduces a novel linear-time inference algorithm for long convolution architectures like SGConv and Hyena. These models typically have a quadratic inference complexity with respect to the sequence length N, which can be prohibitive for large-scale applications.

The key insight behind the proposed approach is a clever partitioning and precomputation strategy for the contributions to the convolution of future outputs. This allows the inference complexity to be reduced to O(N log^2 N). The paper provides extensive experimental evidence demonstrating the acceleration of inference achieved by this new algorithm.

**Strengths:**

* As far as  I know, the interpolation perspective presented in this paper is original and inspiring. The writing is exceptionally clear, and Figure 1 has been extremely helpful in understanding the proposed method.

* The algorithm introduced in the paper largely solves the long-standing problem of quadratic inference complexity for long convolution models like SGConv and Hyena. This has been a significant bottleneck for the practical deployment of these architectures (note that there are also other long conv architectures that do not suffer from this, please see below).

**Weaknesses:**

This is a good paper and there is no much weakness to say about its methodology. However, I find the significance of the work depends on a line of work on long convolution architectures that the authors unfortunately have not discussed or compared.

Long convolution kernels can be contructed from smaller convolutions in a tree style hierarchical dilations such as those in WaveNet [1]. Recently, people have shown that these architectures, with nonlinearities removed and weight sharing, can be interpreted as having a wavelet based state and are competitive with SSM and long convs on sequence modeling benchmarks [2].

Crucially, these dilated convolution architectures also support linear-time inference, as detailed in [3], by maintaining a cache per layer. This allows for efficient inference without the need for techniques like FFT that are required for standard long convolutions.

Given these relevant prior results, I believe a comparison of the proposed approach to these dilated convolution models and their linear-time inference capabilities would greatly strengthen the paper. These alternative architectures offer a potentially simpler approach. As a reader of this paper, it would be nice to know when it is preferable to have a less structured long conv combined with the inference acceleration presented here.

References:

[1] Van Den Oord, Aaron, et al. "Wavenet: A generative model for raw audio." arXiv preprint arXiv:1609.03499 12 (2016).

[2] Shi, Jiaxin, Ke Alexander Wang, and Emily Fox. "Sequence modeling with multiresolution convolutional memory." International Conference on Machine Learning. PMLR, 2023.

[3] Paine, Tom Le, et al. "Fast wavenet generation algorithm." arXiv preprint arXiv:1611.09482 (2016).

**Questions:**

Please see my question above.

---

> ### Author Response · Authors · 2024-11-24
>
> Thank you for acknowledging the presentation and contribution of the paper, as well as for pointing out to the very useful references.
>
> We have mentioned the multiresconv paper and how it has a linear time inference option in Appendix A.2. Granted that structure of a convolutional filter, this inference approach will always be preferable to ours efficiency-wise; we have however added more points about why having a broadly applicable fast LCSM inference method is valuable and can be more valuable in the future. This includes emphasizing the data-dependent setting which we consider a potentially relevant open problem.
>
> On a similar note, we have also added a more nuanced discussion about how our method fits the bigger picture (the whole of Appendix A, with an in-depth analysis of the relationship between LCSMs and SSMs) and the tradeoff between computational efficiency and structural exploitation.

---

> > ### Comment · Reviewer_zv8x · 2024-11-24
> >
> > Hi,
> >
> > Thanks for revising the paper. While the added paragraph better contextualizes this work's position, in general I don't think it is good practice to relegate such closely related research to the appendix with rarely a mention in the paper. This will surely impact readers' ability to make informed decisions – particularly when weighing this approach against simpler alternatives (ofc there is a tradeoff and they have to decide whether they want the dilated conv structure baked in).
> >
> > Discussing this point earlier wouldn't hurt the scientific value of this work in any way but rather helps to identify the optimal use case of the proposed method -- at least this can be done when discussing the need of inference acceleration of LCSMs.

---

> > > ### Author Response · Authors · 2024-12-03
> > >
> > > We agree with the point. The relegation to the appendix was done simply in the interest of space. We will do our best to get as much of the relevant contextual discussion back up to the main paper in the camera-ready version.

---

### Author Response · Authors · 2024-11-23
**First Revision (including vastly improved experiments)**

We provide an intermediate revision of our paper.

The primary experimental improvement we introduce is the use of CUDA Graphs to address CPU overhead in non-mixer components during inference. In both synthetic settings and the Hyena model, these components primarily deal with only current token being generated, where GPU compute time is often shorter than kernel dispatch time. While this overhead is typically hidden by the mixer’s GPU compute time, Flash Inference reduces mixer times substantially (up to $120\times$), exposing the CPU overhead and decreasing end-to-end speed-ups. Lazy execution also exposes this overhead due to its faster performance compared to Eager.
CUDA Graphs resolve this by recording all kernel dispatches for a single token generation into a graph and replaying it for subsequent tokens, dispatching all non-mixer kernels simultaneously. This significantly reduces CPU overhead, making it constant for Flash Inference and its baselines. As a result, Flash Inference achieves up to $120\times$ speed-ups in the mixer and up to $7.8\times$ in end-to-end Hyena runs. A more detailed discussion can be found in Appendix D2 (CPU Overhead: The silent performance killer).

Regarding the points of the reviewers:
- we are currently working on adding a more comprehensive discussion on related techniques and an extended literature review.
- we will also do our best to perform experiments comparing our method to inference for LTI SSMs of various dimensions.

We tried to address every other comment (and referenced this in the corresponding individual rebuttals) except for the ones related to these 2 points, as follows:
- R2W1: Modified lines 108 and 198-199 to emphasize that what we consider a (strict) subset of LCSMs is not SSMs but rather low-dimensional SSMs. We’ll do more about this.
- R2W2: We dropped the references to Hyena at lines 17, 43, 213 and 530 - now it is only referenced for experiments purposes and as an illustrative example at line 165,
- R3W1: need an algorithm
- R3W2: we added an explicit definition of $\tau$ just before Lemma 1 rather than within it. We also mentioned the other 3 implementations in the Appendix (under “More Implementation Details”).
- R3Q1: We have added a mention to Contribution 1 that Laughing Hyena is an approximation rather than exact inference.
- R3Q2: We changed the confusing sentence (former line 233)
- R4W2: added a clear mention to the extra factor of 2 for data-dependent filters at line 214.
- R4[W3/Q4], R5Q2: we added a discussion on causality in Appendix “Framework Adaptations”
- R5W3.2: we added a longer discussion on the connection between across-layer parallelization and memory bandwidth in Appendix “Discussion on Memory”
- R5Q2: we added a discussion on hybrid architectures in Appendix “Framework Adaptations”
- R6W3/Q3: we added comprehensive and easily inspectable data on all our runs (21 experiments) in Appendix Further Experimental Results

---

### Author Response · Authors · 2024-11-24
**Second Revision (added an extensive contextual discussion)**

We have fully updated our background review: updated lines 103-112, 197-199 as well as wrote appendix A on an in-depth contextual discussion. Particularly:
- We changed the perspective to treat LCSMs and SSMs equally and only when relevant for computational reasons distinguish low-dimensional SSMs (as suggested by Reviewer 2)
  - Thus, also emphasized our applicability to SSMs as well. Here, theory predicts that our method will be faster for state dimensions of $\Omega(\log^2(L))$ - however, there is a memory cost associated with it. We are doing our best to get experiments confirming this done in due time.
- Besides SSMs, also commented on how multiresolution convolutions represent another structured class of convolutions where better inference methods exist, as pointed by Reviewer 1.
- Added more comments on the distinction between prefilling and autoregressive generation.

---

### Meta-Review · Area_Chair_idfV · 2024-12-23

**Metareview:**

This paper proposes a method for speeding up long convolution sequence models exact inference to quasilinear time. The paper provides extensive experimental evidence demonstrating the inference benefits of this new algorithm. The reviews for the paper were mostly positive with reviewers highlighting the significance of the work and novelty. I recommend acceptance.

**Additional Comments On Reviewer Discussion:**

The authors did a great job during the rebuttal addressing most concerns raised by the reviewers.

---

### Decision · Program_Chairs · 2025-01-22

Accept (Poster)